# Mixed-Timestep Spiking Neural Networks with Temporal Alignment for Ultra-Low Latency Conversion

## Abstract

Spiking Neural Networks (SNNs) are intrinsically energy-efficient. However, most existing models enforce a uniform time-step across all layers, which limits flexibility and degrades performance under low-latency inference. To address this limitation, we propose *Mixed-Timestep Spiking Neural Networks* (MT-SNNs), a paradigm in which each layer operates with an optimally selected time-step, thereby enabling the joint optimization of accuracy and latency. Within MT-SNNs, we develop a quantization-aware conversion framework that maps a pre-trained ANN to a mixed-timestep SNN. Specifically, we first establish an equivalence principle between activation bit-width and time-steps: the SNN time-step $T$ can be theoretically approximated by the activation quantization level $2^n$ in the source ANN. Based on this theory, different activation bit-widths of ANNs can be layer-wisely mapped to the corresponding time-steps of SNNs. Then, we jointly optimize the quantization levels and firing thresholds to obtain an optimal parameter combination, where the overall $T$ is minimized. As a result, a optimized accuracy-latency trade-off is achieved. Finally, we identify a temporal dimension mismatch issue in MT-SNNs and propose a temporal alignment scheme to address this issue, ensuring proper propagation of activations across layers. Extensive experiments on CIFAR-10, ImageNet-1k, CIFAR10-DVS and DVS-Gesture demonstrate the effectiveness of our approach. On ImageNet-1K, our MT-SNNs achieve 73.63% top-1 accuracy with only 4.88 time-steps, advancing the state of the art.

## 1 Introduction

Spiking Neural Networks (SNNs), regarded as the third generation of neural networks, are inspired by the way biological neurons transmit information via discrete spike trains (Maass, 1997; Izhikevich, 2003; Ghosh-Dastidar & Adeli, 2009). SNNs leverage event-driven computation and temporal dynamics, offering biological plausibility and higher energy efficiency.

Despite these advantages, training SNNs remains challenging due to the non-differentiable nature of spike functions. To address this, surrogate-gradient techniques approximate the gradients of spike activations, enabling enabling the directly training of SNNs with competitive accuracy (Lee et al., 2016; Neftci et al., 2019; Wu et al., 2018b; Lee et al., 2020; Fang et al., 2021; Deng et al., 2022; Fang et al., 2024; Huang et al., 2024). Another widely used approach is ANN-SNN conversion, which circumvents training difficulties by mapping ANN activations to SNN firing rates under rate coding (Cao et al., 2015; Diehl et al., 2015; Rueckauer et al., 2016; Sengupta et al., 2019; Li et al., 2021; Bu et al., 2023; Hao et al., 2023a; Wu et al., 2024; Wang et al., 2025). While effective, the conversion method typically requires long simulation time-steps to achieve high accuracy, resulting in increased inference latency and computational overhead.

Both directly training and conversion methods usually adopt a uniform time-step configuration, where all layers operate with the same number of time-steps (Fang et al., 2021; 2024; Huang et al., 2024; Li et al., 2021; Hao et al., 2023a). This paradigm limits the flexibility and leads to suboptimal trade-offs between accuracy and efficiency. Specifically, both training and inference complexity scale as $\mathcal{O}(NT)$, where $N$ is the number of layers and $T$ is the number of time-steps. Reducing $T$ without sacrificing accuracy is crucial for practical deployment.

To overcome this limitation, we inspired from mixed-precision quantization for ANNs, which shows that different layers exhibit varying sensitivity to quantization noise (Liu et al., 2018; Wu et al., 2018a; Dong et al., 2019; Yao et al., 2021; Elthakeb et al., 2020; Habi et al., 2020). We hypothesize that an analogous principle holds for SNNs: different layers may have different sensitivities to time-step reduction. Allocating more time-steps to sensitive layers and fewer to more robust layers can better balance accuracy and latency.

Based on this insight, we propose Mixed-Timestep Spiking Neural Networks (MT-SNNs), a novel paradigm in which each layer operates with an optimal number of time-step. Unlike existing dynamic time-step methods that use early exit strategies to determine inference time-steps based on input samples' entropy (Li et al., 2023a), our MT-SNNs use statically mapped time-steps and avoids additional runtime decision overhead, making it suitable for deployment.

In this work, we present a quantization-aware conversion framework that constructs MT-SNNs from pre-trained ANNs. We identify optimal parameter combinations that minimize the total number of timesteps while preserving accuracy. To bridge the temporal mismatch between layers, we further propose a temporal alignment mechanism that ensures consistent spike propagation, while compensating for residual errors introduced during conversion. Our main contributions are as follows:

- We establish a theoretical equivalence between activation bit-width and time-steps, showing that time-steps $T$ of SNN can be approximated by the activation quantization level $2^n$ in a pre-trained ANN.
- We jointly optimize quantization levels and firing thresholds to obtain an optimal parameter combination that minimizes overall time-steps $T$ while maintaining high accuracy.
- We identify the temporal mismatch issue in mixed-timestep SNNs and introduce a average temporal alignment strategy to ensure consistent spike propagation across layers, while compensating for residual conversion errors.
- Extensive experiments on CIFAR-10, ImageNet-1K, CIFAR10-DVS, and DVS-Gesture demonstrate the advantages of our method. Notably, our MT-SNNs based on ResNet-34 achieve 73.63% top-1 accuracy, setting a state-of-the-art for low-latency SNN conversion.

## 2 RELATED WORK

**Uniform Time-Steps**. The construction of large-scale deep SNNs predominantly follows two paradigms: (i) directly training with surrogate-gradient methods and (ii) ANN-to-SNN conversion from a pre-trained ANN. Specifically, surrogate-gradient approaches mitigate the non-differentiability of the spike function during directly training (Neftci et al., 2019; Fang et al., 2021; Yao et al., 2022; Huang et al., 2024), making it feasible to train deep SNNs. However, because SNN activation tensors intrinsically have a temporal dimension, treating temporal dimension as a learnable parameter can disrupt standard backpropagation and invalidate surrogate-gradient propagation. Consequently, existing surrogate-gradient methods typically adopt a uniform time-step setting, where all layers share the same time-step. In contrast, ANN-to-SNN conversion maps SNN firing rates to ANN activations (Cao et al., 2015; Rueckauer et al., 2016; Han & Roy, 2020; Li et al., 2021; Bu et al., 2023; Oh & Lee, 2024; Hao et al., 2023a; Wang et al., 2025), where the number of time-steps determines the resolution of the firing rate. This paradigm does not inherently require a uniform time-step across layers. However, prior ANN-to-SNN conversion methods have largely overlooked this flexibility and, similar to directly training, enforce a uniform time-step across all layers, leaving considerable optimization potential unexplored.

**Dynamic Time-Steps**. Inspired by early-exit mechanisms in dynamic ANN networks Almahairi et al. (2016); Chen et al. (2020), Li et al. (2023a) built a decision agent that uses time-varying SNN output confidence to determine inference termination. Li et al. (2023b); Li et al. adjusted time-steps via input sample entropy with confidence thresholds to enable early exit for simple inputs, but this introduces additional runtime decision overhead. Du et al. (2025) proposed mixed-timestep training, where networks are split into stages with random time-steps. To address temporal mismatch between layers, they adopt resampling mechanisms, which can cause feature loss or artificial amplification. In this work, we construct MT-SNNs via an ANN-SNN conversion pipeline, which bypasses the difficulties of directly training SNNs with learnable temporal parameters, offering a practical route to fill the gap in mixed-timestep SNNs.

## 3 PRELIMINARIES

### 3.1 NEURON MODEL

In SNNs, the soft-reset Integrate-and-Fire (IF) neuron is a commonly used model (Cao et al., 2015; Han et al., 2020; Han & Roy, 2020), where the membrane potential is updated rather than reset to a fixed value. The membrane potential of neurons in layer $l$ before spiking at time-step $t$ is determined by integrating the post-spiking potential from the previous time-step $t-1$ and the input current received at time $t$:

$$\boldsymbol{U}_t^l = \boldsymbol{V}_{t-1}^l + \boldsymbol{q}_t^l \tag{1}$$

where $\boldsymbol{U}_t^l$ is the membrane potential of layer $l$ before spiking, and $\boldsymbol{V}_{t-1}^l$ represents the membrane potential of neurons in layer $l$ after spiking at time $t-1$. The input current $\boldsymbol{q}_t^l$ is defined as $\boldsymbol{q}_t^l = \boldsymbol{W}^l \boldsymbol{s}_t^{l-1} \theta^{l-1}$, where $\boldsymbol{W}^l \in \mathbb{R}^{N^l \times N^{l-1}}$ denotes the synaptic weight matrix between layer $l-1$ and layer $l$, and $N^{l-1}$ is the number of neurons in the preceding layer $l-1$. When the $\boldsymbol{U}_t^l$ of a neuron exceeds the threshold $\theta^l$, the neuron fires a spike and update the membrane potential:

$$\boldsymbol{V}_t^l = \boldsymbol{U}_t^l - \theta^l \cdot \boldsymbol{s}_t^l \tag{2}$$

where $\theta^l$ is the firing threshold, and $\boldsymbol{s}_t^l$ represents the spike of layer $l$ at time $t$. The spike $\boldsymbol{s}_t^l$ is determined by the indicator function $\mathbf{1}[\cdot]$ (defined as $\mathbf{1}[x \geq 0] = 1$ and $\mathbf{1}[x < 0] = 0$) as follows:

$$\boldsymbol{s}_t^l = \mathbf{1}[\boldsymbol{U}_t^l \geq \theta^l] \tag{3}$$

**Motivation** A key insight for SNNs is that different layers exhibit varying sensitivities to the number of time-steps. Layers that are more sensitive should be allocated more time-steps to mitigate accuracy degradation, whereas less sensitive layers can use fewer time-steps to reduce memory and computational overhead. However, most prior work enforces a uniform time-step across layers. Although increasing the number of time-steps can improve inference accuracy, it also incurs latency to $O(NT)$, where $N$ is the number of layers and $T$ is the number of time-steps, thereby limiting flexibility and performance in low-latency scenarios. Consequently, to minimize inference latency while preserving accuracy, an effective strategy is to assign layer-wise time-steps.

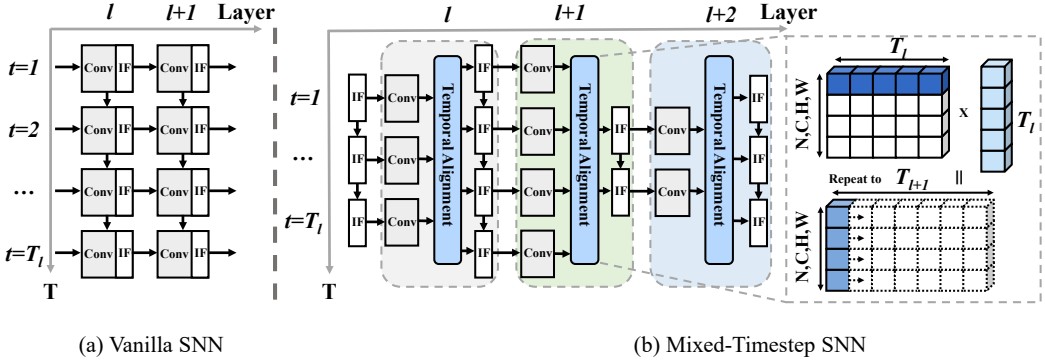

(a) Vanilla SNN      (b) Mixed-Timestep SNN

Figure 1: This figure illustrates the differences between two types of networks: a) Vanilla SNN, where all layers use a uniform time-steps; and b) Mixed-Timestep SNN (MT-SNNs), where time-steps vary across layers.

## 4 METHOD

This section outlines a quantization-aware conversion framework that maps pre-trained ANNs to MT-SNNs. First, a principled relationship between ANN activation bit-width and the number of SNN time-steps is established. Next, it is shown that jointly optimizing quantization levels and firing thresholds yields an optimal accuracy–latency trade-off by minimizing the time-steps. Finally, to resolve inherent temporal dimension mismatch issue under mixed-timestep inference, a temporal alignment mechanism is proposed to ensure consistent inter-layer activation propagation.

## 4.1 EQUIVALENCE RELATIONSHIP BETWEEN ANN AND SNN

**Theorem 1.** *To achieve lossless conversion between a pre-trained ANN and an SNN (i.e., $\boldsymbol{a}^l = \boldsymbol{\phi}^l$), under the premise of eliminating residual membrane potential errors, the time-step $T^l$ of the SNN and the activation quantization level $2^{n^l}$ of the ANN should satisfy the relationship:*

$$T_{SNN} \approx 2^n_{ANN} \tag{4}$$

*proof.* For ANN, the activation quantization operation $\mathcal{Q}(\boldsymbol{a}^l; \boldsymbol{\eta}^l)$ maps $\boldsymbol{a}^l \in \mathbb{R}^d$ to discrete values, where $\boldsymbol{a}^l$ is the activation of the $l$-th layer adn $d$ represents the number of neuron. Here, $\boldsymbol{\eta}^l = [n^l, s^l, \gamma^l]^T$ is the set of trainable parameters for the $l$-th layer (Uhlich et al., 2019), where $n^l \in \mathbb{N}$ is the quantization bit-width, $s^l \in \mathbb{R}$ is the quantization step size, $\gamma^l \in \mathbb{R}$ is the clipping threshold, and the three parameters in $\boldsymbol{\eta}^l$ satisfy the inherent constraint $s^l = \frac{\gamma^l}{2^{n^l}}$. Therefore, the activation quantization formula for ANNs is as follows:

$$\boldsymbol{a}^l = \frac{\gamma^l}{2^{n^l}} \cdot \text{clip}\left( \mathcal{Q}\left( \boldsymbol{W}^l \boldsymbol{a}^{l-1} \cdot \frac{2^{n^l}}{\gamma^l} \right); 0, 2^{n^l} \right) \tag{5}$$

Here, the $\text{clip}(\cdot)$ operation restricts the activation values to the range $[0, 2^{n^l}]$, while $\mathcal{Q}(\cdot)$ denotes rounding quantization, which maps $\boldsymbol{a}^l \in \mathbb{R}^d$ to integer quantization levels and $\boldsymbol{W}^l$ denotes the weights between the $(l-1)$-th layer and the $l$-th layer.

To establish the corresponding expression for SNNs and further derive the equivalence relationship with ANNs, we first start from the membrane potential dynamics of SNNs. Substitute equation 1 into equation 2 to obtain the recursive form $\boldsymbol{V}_t^l = \boldsymbol{V}_{t-1}^l + \boldsymbol{q}_t^l - \theta^l \cdot \boldsymbol{s}_t^l$ and summing this recursive equation over $T^l$ time-steps yields:

$$\boldsymbol{V}_{T^l}^l = \boldsymbol{V}_0^l + \sum_{t=1}^{T^l} \boldsymbol{q}_t^l - \theta^l \cdot \boldsymbol{N}^l \tag{6}$$

where $\boldsymbol{V}_0^l \in \mathbb{R}^d$ is the initial membrane potential of the $l$-th layer, $\boldsymbol{V}_{T^l}^l \in \mathbb{R}^d$ is the residual membrane potential and $\boldsymbol{N}^l = \sum_{t=1}^{T^l} \boldsymbol{s}_t^l \in \{0, 1, \dots, T^l\}^d$ denotes the spike count of the $l$-th layer over $T^l$ time-steps. Define the scaled spike rate as $\boldsymbol{\phi}^l = \frac{\theta^l}{T^l} \cdot \sum_{t=1}^{T^l} \boldsymbol{s}_t^l$, combined with the input current of SNNs $\boldsymbol{q}_t^l = \boldsymbol{W}^l \boldsymbol{s}_t^{l-1} \theta^{l-1}$ rearrange equation 6 and divide both sides by $T^l$. This leads to the expression for the scaled spike firing rate of the $l$-th layer in the SNN:

$$\boldsymbol{\phi}^l = \boldsymbol{W}^l \boldsymbol{\phi}^{l-1} + \frac{\boldsymbol{V}_0^l - \boldsymbol{V}_{T^l}^l}{T^l} \tag{7}$$

To align the form of $\boldsymbol{\phi}^l$ with the ANN activation $\boldsymbol{a}^l$ (equation 5), substitute $\boldsymbol{N}^l = \frac{T^l}{\theta^l} \boldsymbol{\phi}^l$ into equation 6 and rearrange to solve for $\boldsymbol{N}^l$. Considering the integer nature of spike counts ($\boldsymbol{N}^l \in \mathbb{N}^d$) and the range constraint of firing counts ($0 \le N_i^l \le T^l$ for the $i$-th neuron), the spike firing count $\boldsymbol{N}^l$ can be rewritten as:

$$\boldsymbol{N}^l = \text{clip}\left( \mathcal{Q}\left( \frac{\boldsymbol{W}^l \boldsymbol{\phi}^{l-1} \cdot T^l}{\theta^l} + \boldsymbol{\delta}^l \right); 0, T^l \right) \tag{8}$$

where $\boldsymbol{\delta}^l = \frac{\boldsymbol{V}_0^l - \boldsymbol{V}_{T^l}^l}{\theta^l}$ denotes the residual membrane potential error term. Substitute equation 8 back into the definition of $\boldsymbol{\phi}^l = \frac{\theta^l}{T^l} \cdot \boldsymbol{N}^l$, yielding the final expression for the scaled spike firing rate of the $l$-th layer in the SNN:

$$\boldsymbol{\phi}^l = \frac{\theta^l}{T^l} \cdot \text{clip}\left( \mathcal{Q}\left( \frac{\boldsymbol{W}^l \boldsymbol{\phi}^{l-1} \cdot T^l}{\theta^l} + \boldsymbol{\delta}^l \right); 0, T^l \right) \tag{9}$$

The goal of ANN-SNN conversion is to ensure that the scaled spike firing rate of the SNN is consistent with the activation of the ANN. By comparing equation 5 and equation 9 both consist of a scaling factor $\times$ clipped quantization term structure. To achieve $\boldsymbol{\phi}^l = \boldsymbol{a}^l$, the following parameter matching conditions must be satisfied: $\theta^l = \gamma^l$, $T^l = 2^{n^l}$, and $\boldsymbol{\delta}^l = 0$. It should be noted that residual errors arising from $\boldsymbol{\delta}^l \neq 0$ are inevitable. Existing studies have proposed multiple strategies to mitigate such errors, such as weight scaling (Sengupta et al., 2019; Li et al., 2021) and threshold balancing (Diehl et al., 2015). We propose the **average temporal alignment** method in Section 4.3, which enables $\boldsymbol{V}_0^l = \boldsymbol{V}_{T^l}^l$, thereby ensuring $\boldsymbol{\delta}^l = 0$ with the proof provided in A.1.

## 4.2 OPTIMAL PARAMETER COMBINATION

In the proposed conversion framework, the Straight-Through Estimator (STE) (Bengio et al., 2013) is adopted to approximate the gradient of the quantization operation. This approximation enables end-to-end training of the quantization parameter set $\boldsymbol{\eta}^l$ for the conversion task from pre-trained ANNs to MT-SNNs. End-to-end training allows the model to learn optimal per-layer quantization configurations while preserving the accuracy of the pre-trained ANNs. Notably, the learned ANN quantization configurations can be directly mapped to the time-steps of SNNs by the equivalence relationship established in **Theorem 1**. In this section, we analyze gradient flow optimization char-

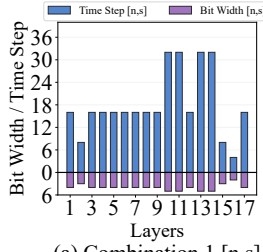 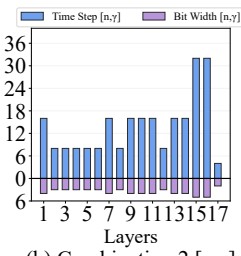 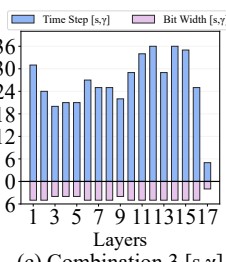 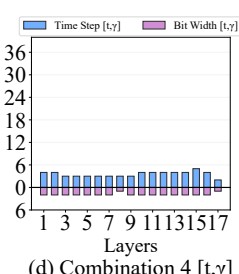

(a) Combination 1 [n,s]  (b) Combination 2 [n,γ]  (c) Combination 3 [s,γ]  (d) Combination 4 [t,γ]

Figure 2: The quantization bit-width in ResNet-18 for ANNs and the corresponding time-step for SNNs on the CIFAR-100 dataset.

acteristics of different parameter combinations from the optimization landscape perspective, demonstrating the superiority of the optimal combination $[t^l, \gamma^l]^T$ in inference latency. For the quantization operation $\mathcal{Q}(\boldsymbol{a}^l; \boldsymbol{\eta}^l)$, the parameter set $\boldsymbol{\eta}^l$ satisfies the inherent constraint $s^l = \frac{\gamma^l}{2^{n^l}}$, which implies that once any two parameters are determined, the third is uniquely fixed. The feasible four parameter combinations include: $[n^l, s^l]^T$, $[n^l, \gamma^l]^T$, $[s^l, \gamma^l]^T$, and $[t^l, \gamma^l]^T$ (where $t^l = 2^{n^l}$ treats the quantization level). To jointly optimize accuracy and latency, a multi-objective loss function is designed as follows:

$$\mathcal{J}(\boldsymbol{\eta}) = \mathcal{L}\left(\mathcal{Q}\left(\boldsymbol{a}^l; \boldsymbol{\eta}^l\right)\right) + \lambda \sum_l t^l \tag{10}$$

Where, $\mathcal{L}$ denotes the cross-entropy classification loss, $\lambda > 0$ is the regularization coefficient that balances the trade-off between classification loss and latency constraints, and $t^l$ represents the equivalent time-step of the $l$-th layer under four different parameter combinations. Let $\boldsymbol{g}^l = \frac{\partial \mathcal{L}}{\partial \boldsymbol{a}^l}$ denote the gradient of the classification loss with respect to the $l$-th layer activation $\boldsymbol{a}^l$. The gradient calculation results for the four parameter combinations are presented below. For ease of derivation, a mask function $\boldsymbol{I}(\cdot)$ is introduced, which operates element-wise on input vectors and outputs a 0-1 mask matrix of the same dimension. This function is used to handle the clip($\cdot$) operation in quantization and simplify gradient expressions. Specifically, the clip($x; 0, C$) (where C denotes the upper bound of clipping) operation can be rewritten as a linear combination via the mask function: clip($x; 0, C) = x \cdot \boldsymbol{I}(0 \leq x \leq C) + C \cdot \boldsymbol{I}(x > C)$, which unifies gradient expressions across different parameter combinations.

**Combination 1** $[n^l, s^l]^T$: For bit-width $n^l$ and step size $s^l$, the gradients of total loss $\mathcal{J}$ are:

$$\frac{\partial \mathcal{J}}{\partial n^l} = 2^{n^l} \ln(2) \left[\boldsymbol{g}^l \cdot s^l \cdot \boldsymbol{I}(\hat{\boldsymbol{z}}^l > 2^{n^l}) + \lambda\right] \tag{11}$$

$$\frac{\partial \mathcal{J}}{\partial s^l} = \boldsymbol{g}^l \cdot \left[(\hat{\boldsymbol{z}}^l - \frac{\boldsymbol{z}^l}{s^l}) \odot \boldsymbol{I}(\hat{\boldsymbol{z}}^l \leq 2^{n^l}) + 2^{n^l} \cdot \boldsymbol{I}(\hat{\boldsymbol{z}}^l > 2^{n^l})\right] \tag{12}$$

where $\hat{\boldsymbol{z}}^l = \mathcal{Q}(\frac{\boldsymbol{z}^l}{s^l})$, $\odot$ is element-wise multiplication, and the equivalent time-step is $t^l = 2^{n^l}$.

**Combination 2** $[n^l, \gamma^l]^T$: For bit-width $n^l$ and clipping threshold $\gamma^l$, the gradients of loss $\mathcal{J}$ are:

$$\frac{\partial \mathcal{J}}{\partial n^l} = \ln(2) \left[\boldsymbol{g}^l \cdot \boldsymbol{G}_n^l + \lambda \cdot 2^{n^l}\right] \tag{13}$$

$$\frac{\partial \mathcal{J}}{\partial \gamma^l} = \boldsymbol{g}^l \cdot \left[(s^l \hat{\boldsymbol{u}}^l - \frac{\boldsymbol{z}^l}{\gamma^l}) \odot \boldsymbol{I}(\hat{\boldsymbol{u}}^l \leq 2^{n^l}) + \boldsymbol{I}(\hat{\boldsymbol{u}}^l > 2^{n^l})\right] \tag{14}$$

where $\hat{\boldsymbol{u}}^l = \mathcal{Q}(\boldsymbol{z}^l \cdot \frac{2^{n^l}}{\gamma^l})$, $\boldsymbol{G}_n^l = (\boldsymbol{z}^l - s^l\hat{\boldsymbol{u}}^l) \odot \boldsymbol{I}(\hat{\boldsymbol{u}}^l \leq 2^{n^l}) + \gamma^l \boldsymbol{I}(\hat{\boldsymbol{u}}^l > 2^{n^l})$, and the equivalent inference time-step of the $l$-th layer in the SNN is $t^l = 2^{n^l}$.

**Combination 3** $[s^l, \gamma^l]^T$: For step size $s^l$ and clipping threshold $\gamma^l$, the gradient expressions are:

$$\frac{\partial \mathcal{J}}{\partial s^l} = \boldsymbol{g}^l \cdot (\hat{\boldsymbol{z}}^l - \frac{\boldsymbol{z}^l}{s^l}) \odot \boldsymbol{I}(\hat{\boldsymbol{z}}^l \leq \frac{\gamma^l}{s^l}) - \lambda \frac{\gamma^l}{(s^l)^2} \tag{15}$$

$$\frac{\partial \mathcal{J}}{\partial \gamma^l} = \boldsymbol{g}^l \cdot \boldsymbol{I}(\hat{\boldsymbol{z}}^l > \frac{\gamma^l}{s^l}) + \frac{\lambda}{s^l} \tag{16}$$

where the equivalent inference time-step of the $l$-th layer in the target SNN is $t^l = \frac{\gamma^l}{s^l}$.

**Combination 4** $[t^l, \gamma^l]^T$: For time-step $t^l$ and clipping threshold $\gamma^l$, the gradients of loss $\mathcal{J}$ are:

$$\frac{\partial \mathcal{J}}{\partial t^l} = \boldsymbol{g}^l \cdot \boldsymbol{G}_t^l + \lambda \tag{17}$$

$$\frac{\partial \mathcal{J}}{\partial \gamma^l} = \boldsymbol{g}^l \cdot \left[ (\frac{\hat{\boldsymbol{v}}^l}{t^l} - \frac{\boldsymbol{z}^l}{\gamma^l}) \odot \boldsymbol{I}(\hat{\boldsymbol{v}}^l \leq t^l) + \boldsymbol{I}(\hat{\boldsymbol{v}}^l > t^l) \right] \tag{18}$$

where $\hat{\boldsymbol{v}}^l = \mathcal{Q}(\boldsymbol{z}^l \cdot \frac{t^l}{\gamma^l})$, $\boldsymbol{G}_t^l = (\frac{\boldsymbol{z}^l}{t^l} - \frac{\gamma^l\hat{\boldsymbol{v}}^l}{(t^l)^2}) \odot \boldsymbol{I}(\hat{\boldsymbol{v}}^l \leq t^l) + \frac{\gamma^l}{t^l} \boldsymbol{I}(\hat{\boldsymbol{v}}^l > t^l)$ and $\boldsymbol{t}^l$ is the quantization level, which directly corresponds to the inference time-step of the $l$-th layer in the target SNN.

From the above gradient expressions, significant differences in numerical stability among four parameter combinations can be observed. **Combinations 1 and 2** include the exponential term $2^{n^l}$, which may lead to gradient explosion when the bit-width $n^l$ is large. In particular, the factor $2^{n^l} \ln(2)$ in equation 11 and equation 13 is prone to causing numerical instability during the training of deep networks. **Combination 3** avoids exponential and logarithmic operations, resulting in relatively concise gradient expressions. However, the regularization term $\frac{\gamma^l}{(s^l)^2}$ in equation 15 remains problematic: when the quantization step size $s^l$ is extremely small, this term can become excessively large, potentially causing numerical fluctuations. **Combination 4** exhibits optimal numerical stability, with the reg-

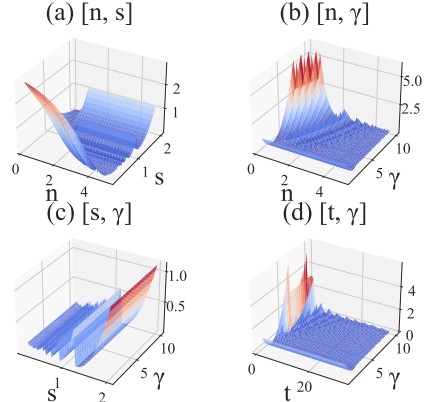

Figure 3: Loss surfaces of four different parameter combinations

ularization term in the time-step gradient being a constant $\lambda$, completely avoiding exponential, logarithmic, and high-order fractional terms, making its gradient expression the most stable. As visualized in Figure 3 **Combinations 4** yields the smoothest landscape, stable gradients, and fastest convergence.

### 4.3 TEMPORAL ALIGNMENT

In MT-SNNs, different layers operate at distinct temporal resolutions, leading to potential temporal mismatch between consecutive layers. Suppose the output activation of the $l$-th layer is $\boldsymbol{S}^l$ with a shape of $\boldsymbol{S}^l \in \mathbb{R}^{B \times T^l \times C^l \times H^l \times W^l}$, where $B$ is the batch size, $T^l$ is the time-step of the $l$-th layer, $C^l$ is the number of channels, and $H^l$ and $W^l$ denote the height and width of the feature map in the $l$-th layer, respectively. Let $T^{l+1}$ represent the time-steps required for the $(l+1)$-th layer. Since $T^l \neq T^{l+1}$ in most cases, the output spike tensor $\boldsymbol{S}^l$ needs temporal alignment to match $T^{l+1}$. We define temporal alignment as the operation described below:

**Definition.** *Temporal alignment refers to the operation that enforces consistency in the temporal dimension between layers with different time-steps.*

$$\hat{\boldsymbol{S}}^{l+1} = f(\boldsymbol{S}^l, T^{l+1}) \tag{19}$$

Where, $\hat{\boldsymbol{S}}^{l+1}$ is the expected input of the $(l+1)$-th layer, and $f(\cdot)$ denotes the temporal alignment function, which adjusts $\boldsymbol{S}^l$ temporal dimension from $T^l$ to $T^{l+1}$. As illustrated in Figure 1, temporal alignment is applied to $\boldsymbol{S}^l$ before input to the $(l+1)$-th layer when $T^l \neq T^{l+1}$.

To address this, we propose the **average temporal alignment** method, which consists two steps. First, we average $\boldsymbol{S}^l$ over the temporal dimension $T^l$ to obtain a time-step-independent feature tensor $\bar{\boldsymbol{S}}^l \in \mathbb{R}^{B \times C^l \times H^l \times W^l}$, where $\boldsymbol{S}^l[t]$ is the activation at $t$ time-step. Second, we replicate $\bar{\boldsymbol{S}}^l$ along the temporal dimension to match the required length $T^{l+1}$, yielding the aligned tensor $\hat{\boldsymbol{S}}^{l+1} \in \mathbb{R}^{T^{l+1} \times B \times C^l \times H^l \times W^l}$. This process is formally defined as:

$$\bar{\boldsymbol{S}}^l = \frac{1}{T^l} \sum_{t=1}^{T^l} \boldsymbol{S}^l[t], \quad \hat{\boldsymbol{S}}^{l+1} = \bar{\boldsymbol{S}}^l \otimes \mathbf{1}_{T^{l+1},1} \tag{20}$$

where $\mathbf{1}_{T^{l+1},1}$ is a ones tensor of length $T^{l+1}$, and it is used to replicate $\bar{\boldsymbol{S}}^l$ across all time-steps.

## 5 Experiments

This section first compares MT-SNNs against state-of-the-art conversion and dynamic time-step methods, demonstrating an optimal accuracy–latency trade-off. Next, ablations over four parameter combinations identify the combination four as optimal. Subsequently, ablation studies on the temporal alignment module highlight the average temporal alignment strategy as the most effective strategy. Finally, a computational overhead analysis of the average temporal alignment module shows that the temporal alignment maintains low overhead.

### 5.1 Comparison with Previous Works.

We evaluate the proposed method against state-of-the-art conversion approaches on ImageNet-1K dataset. We primarily focus on SNN accuracy and average inference time-steps $\bar{T} = \frac{\sum_{l=1}^L T^l}{L}$, aiming to demonstrate that our method can substantially reduce inference latency while maintaining high accuracy.

Table 1: Comparison between our method and previous works on ImageNet-1k dataset.

| Method | VGG-16 | | | ResNet-34 | | |
|---|---|---|---|---|---|---|
| | ANN | T | Accuracy(%) | ANN | T | Accuracy(%) |
| Calibration(Li et al., 2021) | 75.36 | 16, 32 | 43.99, 62.14 | 75.66 | 16, 32 | 34.91, 61.43 |
| SlipReLU(Jiang et al., 2023) | 71.99 | 16, 32 | 51.54, 67.48 | 75.08 | 16, 32 | 43.76, 66.61 |
| QCFS(Bu et al., 2023) | 74.29 | 16,32 | 50.97, 68.47 | 74.32 | 16, 32 | 59.35, 69.37 |
| SRP*(Hao et al., 2023b) | 74.29 | 4, 8 | 66.47, 68.37 | 74.32 | 4, 8 | 66.71, 67.62 |
| FTBC(Wu et al., 2024) | 75.36 | 8, 16 | 64.20, 71.19 | 75.66 | 8, 16 | 38.55, 60.68 |
| AdaFire(Wang et al., 2025) | 75.36 | 8, 16 | 73.53, 74.25 | 75.66 | 8, 16 | 72.96, 73.85 |
| **ours (MT-SNNs)** | 73.61 | **4.93** | **73.12** | 74.01 | **4.88** | **73.63** |

Note: SRP* requires $\tau$ time-steps to be executed before actual inference, so the actual inference time should be $T + \tau$. In the original paper, $\tau = 14$ for VGG-16 and $\tau = 8$ for ResNet-34.

**Results on ImageNet-1K.** As shown in Table 1, for the two architectures VGG-16 and ResNet-34, the proposed method (MT-SNNs) achieves accuracies of 73.12% and 73.63% with only 4.93 and 4.88 time-steps, respectively. Compared with AdaFire, which achieves accuracies of 73.53% and 74.25% with 8 and 16 time-steps on VGG-16, and 72.96% and 73.85% with the same time-steps on ResNet-34, respectively, our MT-SNNs maintains nearly lossless accuracy while reducing the number of time-steps by 50%.

For VGG-16, compared with the SRP method, which achieves an accuracy of 66.47% with 4 time-steps, our MT-SNNs outperforms SRP by 6.87% with 4.93 time-steps. For ResNet-34, SRP achieves an accuracy of 66.71% with 4 time-steps, and our MT-SNNs outperforms SRP by 6.92% with 4.88 time-steps. Additionally, SRP requires $\tau$ time-steps to be executed before actual inference; thus, the

actual inference time should be $T + \tau$. According to the original paper of SRP, $\tau$ is set to 14 for VGG-16 and 8 for ResNet-34. As shown in Appendix Tables 5 and 6, the experimental results on the CIFAR-10 and DVS datasets also outperform those of previous works.

**Comparison with Dynamic Timestep Methods.** To validate the effectiveness of our conversion framework in dynamic temporal modeling, we compare the proposed MT-SNNs against state-of-the-art dynamic time-step SNNs on ImageNet-1K with ResNet-34, as shown in Table 2. "Dir. Train." denotes directly training approaches, and "Conv." refers to conversion-based methods. Compared to directly training methods such as SEENN-I and MTT,

Table 2: Comparison our method and previous dynamic time-step works on ImageNet-1k with ResNet-34.

| Method | Type | T | SNN |
|---|---|---|---|
| SEENN-I (Li et al., 2023b) | Dir.Train. | 2.28, 3.38 | 63.65, 64.66 |
| SEENN-I (Li et al., 2023b) | Conv. | 23.47, 29.53 | 70.18, 71.84 |
| MTT (Du et al., 2025) | Dir.Train. | 4, 6 | 67.54, 68.34 |
| AdaFire (Wang et al., 2025) | Conv. | 8, 16 | 72.96, 73.85 |
| **ours (MT-SNNs)** | Conv. | **4.88** | **73.63** |

our MT-SNNs achieves a superior accuracy-latency trade-off. While SEENN-I and MTT only attain 64.66% (with 3.38 time-steps) and 68.34% (with 6 time-steps), respectively, our MT-SNNs reaches 73.63% with merely 4.88 time-steps. Among conversion-based methods, our MT-SNNs also stands out: SEENN-I achieves 71.84% but requires 29.53 time-steps, while AdaFire reaches 73.85% with 16 time-steps. In contrast, our MT-SNNs delivers comparable accuracy with significantly lower latency, which demonstrates its efficiency and scalability.

## 5.2 ABLATION STUDY ON TEMPORAL ALIGNMENT

We proposed the average temporal alignment in Section 4.3. Beyond average, other temporal alignment schemes exist, such as naive upsampling or downsampling adopted in prior dynamic time-step work (Du et al., 2025), where resampling is performed after feature grouping. The key challenge of such resampling lies in segmenting the temporal without introducing feature distortion; however, resampling inevitably incurs feature loss or artificial amplification. To verify the advantage of average temporal alignment, we conducted ablation studies over four alignment strategies. Table 3 reports the results for (**Up+Avg**, **Up+Down**, **Avg+Down**, **Avg+Avg**) on CIFAR-10 and CIFAR-100 with ResNet-18, ResNet-20, and VGG-16. Here, **Avg** denotes average-based alignment, **Up** denotes upsampling, and **Down** denotes downsampling. Key findings are as follows:

Table 3: Temporal Alignment Ablation Study.

| Method | CIFAR-10 / CIFAR-100 | | |
|---|---|---|---|
| | ResNet-18 (%) | ResNet-20 (%) | VGG-16 (%) |
| Up + Avg | 95.31 / 75.50 | 89.52 / 64.86 | 94.45 / 75.70 |
| Up + Down | 94.16 / 71.21 | 87.67 / 55.24 | 91.23 / 60.65 |
| Avg + Down | 94.16 / 75.93 | 89.20 / 62.91 | 91.23 / 74.06 |
| **Avg + Avg** | **95.71 / 77.82** | **91.08 / 65.39** | **94.98 / 75.89** |

(1) **Dominance of Avg+Avg strategy**: Consistent with the theoretical analysis in A.1, **Avg+Avg** achieves the highest accuracy across all backbones and datasets, outperforming other strategies by 1.40%–3.75% on CIFAR-10 and 1.48%–15.24% on CIFAR-100, with the largest gain 15.24% observed on CIFAR-100 with VGG-16. (2) **Superiority of average operations**: Strategies involving average (**Avg+Avg**, **Up+Avg**, **Avg+Down**) outperform **Up+Down**, indicating that average-based alignment better preserves temporal consistency and reduces residual errors compared with discrete resampling. (3) **Generalizability across backbones**: The relative advantage of **Avg+Avg** over other strategies is consistent across ResNet-18, ResNet-20, and VGG-16, demonstrating the generalization. Overall, the **Avg+Avg** alignment scheme offers consistent optimal performance, demonstrating strong generalization across models and datasets.

## 5.3 ABLATION EXPERIMENTS ON FOUR PARAMETER COMBINATIONS

We conduct ablation experiments with ResNet-18 on CIFAR-10, CIFAR-100, and ImageNet-200 to evaluate model accuracy, the average quantization bit-width, and the corresponding time-steps under four parameter combinations. As shown in Figure 4, the purple bar chart denotes the average

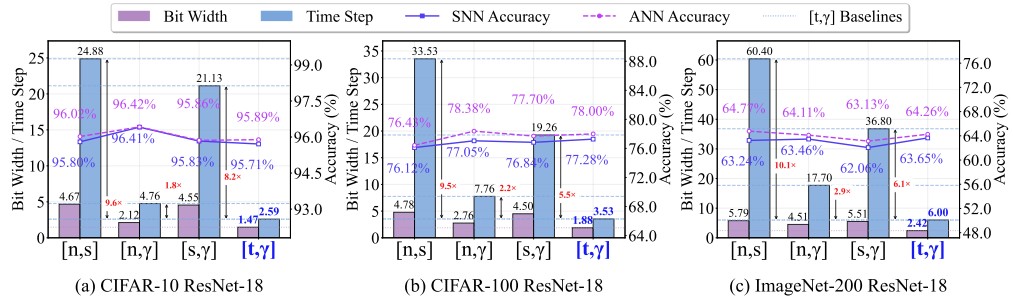

Figure 4: The average bit-width, time-steps, ANN and SNN accuracy under different quantization parameter combination

bit-width $\bar{n} = \frac{\sum_{n=1}^{L} n^l}{L}$, and the blue bar chart represents the time-step $\bar{T} = \frac{\sum_{l=1}^{L} T^l}{L}$. **Combination 4** $[t, \gamma]$ achieves the lowest average time-step. On CIFAR-10, it reduces inference latency by $9.6\times$, $1.8\times$, and $8.2\times$ over **Combination 1**, **Combination 2**, and **Combination 3**, respectively, while achieving 95.71% Top-1 accuracy. On CIFAR-100, it yields $9.5\times$, $2.2\times$, and $5.5\times$ latency reductions over the three combinations and attains the highest conversion accuracy of 77.28%. On ImageNet-200, it achieves $10.1\times$, $2.9\times$, and $6.1\times$ latency reductions, with a Top-1 accuracy of 63.65%.

## 5.4 Computational Overhead of Temporal Alignment

To evaluate the overhead of the average temporal alignment operation, we conduct experiments on the ImageNet-1K validation with ResNet-34 and VGG-16. Experiments are run on $4\times$ NVIDIA RTX 4090 GPUs (24 GB each) with a fixed global inference batch size of 64. We report the total wall-clock inference time (**Inf. Time**), the time spent on temporal alignment (**Align. Time**), and its proportion of total inference time. Results are summarized in Table 4.

Table 4: Inference Overhead on ImageNet-1K dataset.

| Architecture | Model | Top-1 Acc. (%) | Avg. T | Inf. Time (s) | Align. Time (s) | Speedup($\uparrow$) |
|---|---|---|---|---|---|---|
| **ResNet-34** | Vanilla-SNN | 69.47 | 32 | 1893.38 | - | $1\times$ |
| | MT-SNNs | 73.63 | 4.88 | 344.44 | 15.39 (4.47%) | $5.49\times$ |
| **VGG-16** | Vanilla-SNN | 68.56 | 32 | 4,465.28 | - | $1\times$ |
| | MT-SNNs | 73.12 | 4.93 | 704.31 | 37.75 (5.36%) | $6.34\times$ |

For ResNet-34, the total inference time of MT-SNNs is 344.44 s, whereas the Vanilla-SNNs (with uniform timesteps) require 1893.38 s, yielding a $5.49\times$ speedup. The temporal-alignment operation takes 15.39 s, accounting for only 4.47% of the total inference time. For VGG-16, the total inference time of MT-SNNs is 704.31 s, compared to 4465.28 s for the Vanilla-SNNs, achieving a $6.34\times$ speedup. The alignment time is 37.75 s, corresponding to a 5.36% overhead. These results indicate that average temporal alignment mechanism introduces a low computational overhead while improving inference speed.

## 6 Conclusion

In this work, we propose Mixed-Timestep Spiking Neural Networks (MT-SNNs). To bypass the difficulties of directly training MT-SNNs, we introduce a quantization-aware conversion framework that bridges quantized ANNs and SNNs with mixed timesteps. We first theoretically establish the equivalence between activation quantization levels of ANN and times-teps of SNN, and then demonstrate that jointly optimizing the quantization level and the firing threshold yields optimal low-latency inference while preserving accuracy. We further identify the temporal mismatch issue in MT-SNNs and propose an average temporal alignment mechanism to address it. Extensive experiments demonstrate the advantages of our MT-SNNs, filling the gap in mixed-timestep SNNs.

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

# A APPENDICES

## A.1 AVERAGE TEMPORAL ALIGNMENT ELIMINATES RESIDUAL ERRORS

In Section 4.1, we established the equivalence relationship between ANNs and SNNs. However, when $\theta^l = \gamma^l$ and $T^l = 2^{n^l}$, equation 5 and equation 9 are not completely equivalent due to the presence of $\boldsymbol{\delta}^l = \frac{\boldsymbol{V}_0^l - \boldsymbol{V}_{T^l}^l}{\theta^l}$, resulting in residual membrane potential error. Below, we prove that **average temporal alignment** is the optimal method among temporal alignment approaches, as it eliminates the residual error $\boldsymbol{\delta}^l = 0$ by ensuring $\boldsymbol{V}_0^l = \boldsymbol{V}_{T^l}^l$.

When the SNN adopts average temporal alignment, the input current $\boldsymbol{q}_t^l$ of the $l$-th layer at each time-step is derived from equation 20, as shown follows:

$$\boldsymbol{q}_t^l = \frac{\theta^{l-1}}{T^l} \cdot \boldsymbol{W}^l \sum_t^{T^{l-1}} \boldsymbol{s}_t^{l-1} \tag{21}$$

Summing the input current over the total time-steps $T^l$ of the $l$-th layer gives: $\sum_{t=1}^{T^l} \boldsymbol{q}_t^l = T^l \cdot \boldsymbol{q}_t^l = \theta^{l-1} \cdot \boldsymbol{W}^l \sum_{t=1}^{T^{l-1}} \boldsymbol{s}_t^{l-1}$. Under the ANN-SNN equivalence premise $\phi^{l-1} = \boldsymbol{a}^{l-1}$, the spike count of the $(l-1)$-th layer $\boldsymbol{N}^{l-1} = \sum_{t=1}^{T^{l-1}} \boldsymbol{s}_t^{l-1}$ is equal to the ANN's quantization level $\boldsymbol{M}^{l-1}$ (i.e., $\boldsymbol{N}^{l-1} = \boldsymbol{M}^{l-1}$). Substituting this into the above equation yields $\sum_{t=1}^{T^l} \boldsymbol{q}_t^l = \theta^{l-1} \cdot \boldsymbol{W}^l \cdot \boldsymbol{M}^{l-1}$.

For ANN-SNN conversion, the parameter matching conditions $\theta^{l-1} = \gamma^{l-1}$ and $T^l = 2^{n^l}$ hold, and the ANN's activation satisfies $\boldsymbol{a}^{l-1} = \frac{\gamma^{l-1}}{2^{n^{l-1}}} \cdot \boldsymbol{M}^{l-1}$ (from equation 5). Combining these, the total input current of the $l$-th layer can be rewritten as $\sum_{t=1}^{T^l} \boldsymbol{q}_t^l = T^l \cdot \boldsymbol{W}^l \cdot \boldsymbol{a}^{l-1} = T^l \cdot \boldsymbol{a}^l = \theta^l \cdot \boldsymbol{M}^l$ where $\boldsymbol{a}^l = \boldsymbol{W}^l \cdot \boldsymbol{a}^{l-1}$ is the activation of the $l$-th layer in the ANN, and $\boldsymbol{M}^l$ is its corresponding quantization level matrix.

Rearranging equation 6 to solve for the spike count $\boldsymbol{N}^l$ gives $\boldsymbol{N}^l = \frac{\boldsymbol{V}_0^l + \sum_{t=1}^{T^l} \boldsymbol{q}_t^l - \boldsymbol{V}_{T^l}^l}{\theta^l}$. Since the membrane potential is soft-reset to $[0, \theta^l)$ after each spike, so the residual membrane potential at the final time-step satisfies $0 \leq \boldsymbol{V}_{T^l}^l < \theta^l$. Substituting this constraint into the equation for $\boldsymbol{N}^l$ yields the range of $\boldsymbol{N}^l \in (\frac{\boldsymbol{V}_0^l + \sum_{t=1}^{T^l} \boldsymbol{q}_t^l - \theta^l}{\theta^l}, \frac{\boldsymbol{V}_0^l + \sum_{t=1}^{T^l} \boldsymbol{q}_t^l}{\theta^l}]$. Since $\boldsymbol{N}^l \in \mathbb{Z}^d$ is an integer, it follows that $\boldsymbol{N}^l = \lfloor \frac{\boldsymbol{V}_0^l + \boldsymbol{Q}_{tot}^l}{\theta^l} \rfloor$. Substituting $\sum_{t=1}^{T^l} \boldsymbol{q}_t^l = \theta^l \cdot \boldsymbol{M}^l$ into the above equation gives $\boldsymbol{N}^l = \lfloor \frac{\boldsymbol{V}_0^l + \theta^l \cdot \boldsymbol{M}^l}{\theta^l} \rfloor$.

In practical SNN inference, we follow the initial membrane potential setting from Bu et al. (Bu et al., 2022; 2023), where $\boldsymbol{V}_0^l = 0.5\theta^l$. Given that $\boldsymbol{M}^l \in \mathbb{Z}^d$, the equation can be simplified to $\boldsymbol{N}^l = \lfloor \frac{0.5\theta^l + \theta^l \cdot \boldsymbol{M}^l}{\theta^l} \rfloor = \lfloor \boldsymbol{M}^l + 0.5 \rfloor = \boldsymbol{M}^l$

Substituting $\sum_{t=1}^{T^l} \boldsymbol{q}_t^l = \theta^l \cdot \boldsymbol{M}^l$ and $\boldsymbol{N}^l = \boldsymbol{M}^l$ back into Equation equation 6 confirms that the residual membrane potential error is eliminated: $\boldsymbol{V}_{T^l}^l - \boldsymbol{V}_0^l = \sum_{t=1}^{T^l} \boldsymbol{q}_t^l - \theta^l \cdot \boldsymbol{N}^l = \theta^l \cdot \boldsymbol{M}^l - \theta^l \cdot \boldsymbol{M}^l = 0$. Thus, $\boldsymbol{V}_0^l = \boldsymbol{V}_{T^l}^l$, which implies $\boldsymbol{\delta}^l = 0$ and satisfies the equivalence condition of Theorem 4.1.

## A.2 EXPERIMENTAL PARAMETER SETTINGS

For training VGG-16, ResNet-18, and ResNet-20 on CIFAR-10 and CIFAR-100 (Krizhevsky et al., 2009): the training batch size is set to 300, and the number of training epochs is 300. The Stochastic Gradient Descent (SGD) optimizer (Bottou, 2012) is adopted, with an initial learning rate of 0.01 for VGG-16 and ResNet-18, and 0.05 for ResNet-20. The regularization coefficient $\lambda$ is set to 1e-6, and a cosine annealing scheduler (Loshchilov & Hutter, 2016) is used; additional configurations include a weight decay of $5 \times 10^{-4}$ and 5 warm-up epochs. For data preprocessing, standard augmentation techniques are applied, including random cropping, Cutout (DeVries & Taylor, 2017), and AutoAugment (Cubuk et al., 2019).

For training ResNet-34 and VGG-16 on ImageNet-1K (Deng et al., 2009): the training batch size is 64, the test batch size is 100, and the number of training epochs is 120. The SGD optimizer (Bottou, 2012) is used, paired with a cosine annealing scheduler (Loshchilov & Hutter, 2016), 5 warm-up epochs, and a weight decay of 1e-4. Mixed-precision training is implemented on 4 NVIDIA RTX 4090 GPUs, and augmentation techniques include Cutout and AutoAugment(Cubuk et al., 2019).

For dynamic vision sensor (DVS) datasets CIFAR10-DVS and DVS-Gesture: CIFAR10-DVS uses a training batch size of 32 and a test batch size of 10, while DVS-Gesture uses a test batch size of 18. Shared parameters across both datasets are as follows: 300 training epochs, the SGD optimizer with a learning rate of 0.1, a cosine annealing scheduler, 5 warm-up epochs, a weight decay of 5e-4, and mixed-precision training. During training, each sample in the datasets is converted to 2 frames, with data augmentation applied from (Hao et al., 2024).

**Analysis of Results on CIFAR-10 Dataset** Table 5 shows that on CIFAR-10, our method achieves SNN accuracies of 94.98%, 95.71%, and 91.08% on VGG-16, ResNet-18, and ResNet-20, respectively, with only 2.20, 2.59, 3.63 time-steps. Compared to QCFS and SlipReLU (4–8 time-steps), it achieves over 50% temporal compression while matching or slightly exceeding top baselines (e.g., SRP at 95.60%). SRP requires $\tau = 4$ pre-inference time-steps, so its actual inference time is $T + \tau$. Additionally, versus high-accuracy methods like SGDND (16–32 time-steps), our method maintains comparable accuracy (0.5% gap) with 85% shorter temporal sequences.

Table 5: Comparison between our method and previous works on CIFAR-10 dataset.

| Architecture | Methods | ANN(%) | T | SNN(%) |
|---|---|---|---|---|
| VGG-16 | QCFS(Bu et al., 2023) | 95.52 | 4, 8 | 93.96, 94.95 |
|  | SRP*(Hao et al., 2023a) | 95.52 | 4, 8 | 95.32, 95.52 |
|  | SlipReLU(Jiang et al., 2023) | 93.02 | 4, 8 | 91.08, 92.26 |
|  | SGDND(Oh & Lee, 2024) | 95.96 | 16, 32 | 81.06, 95.53 |
|  | **ours (MT-SNNs)** | 94.98 | **2.20** | **94.98** |
| ResNet-18 | QCFS(Bu et al., 2023) | 96.04 | 2, 4 | 91.75, 93.83 |
|  | SRP*(Hao et al., 2023a) | 95.64 | 4, 8 | 95.25, 95.60 |
|  | SlipReLU(Jiang et al., 2023) | 94.61 | 2, 4 | 93.97, 94.59 |
|  | SGDND(Oh & Lee, 2024) | 96.82 | 16, 32 | 80.74, 96.29 |
|  | **ours (MT-SNNs)** | 95.89 | **2.59** | **95.71** |
| ResNet-20 | QCFS(Bu et al., 2023) | 91.77 | 4, 8 | 83.75, 89.55 |
|  | SRP*(Hao et al., 2023a) | 91.77 | 4, 8 | 90.51, 90.51 |
|  | SlipReLU(Jiang et al., 2023) | 92.96 | 8, 16 | 86.66, 92.13 |
|  | SGDND(Oh & Lee, 2024) | 96.82 | 16, 32 | 80.74, 96.29 |
|  | **ours (MT-SNNs)** | 91.41 | **3.63** | **91.08** |

## A.3 TEMPORAL CHANNEL RESHAPING FOR DYNAMIC DATA

Neuromorphic datasets include an extra temporal dimension $T$, preventing direct ANN pre-training. We propose two schemes to address this: **TB** (merges $T$ with batch dimension $B$) and **TC** (integrates $T$ into channel dimension $C$), enabling ANN pre-training on such datasets.

We compared training time and memory overhead of **TC**, **TB**, and SNNs using ResNet-18 on DVS-Gesture (batch size 16, single Nvidia 3090ti, 10-epoch average). Figure 5 show **TC** has slightly

Table 6: Comparison between our method and previous works on neuromorphic datasets.

| Datasets | Methods | Models | T | Acc.(%) |
|----------|---------|--------|---|---------|
| DVS-Gesture | PLIF(Fang et al., 2021) | PLIF Net | 20 | 97.57 |
| | KLIF(Jiang & Zhang, 2023) | Modified PLIF Net | 12 | 94.10 |
| | CLIF(Huang et al., 2024) | Spiking-Vgg11 | 20 | 97.92 |
| | **ours (MT-SNNs)** | ResNet-18 | **4.06** | **94.10** |
| CIFAR10-DVS | PLIF(Fang et al., 2021) | PLIF Net | 20 | 74.80 |
| | KLIF(Jiang & Zhang, 2023) | Modified PLIF Net | 15 | 70.90 |
| | CLIF(Huang et al., 2024) | Spiking-Vgg11 | 16 | 79.00 |
| | **ours (MT-SNNs)** | ResNet-18 | **2.0** | **84.40** |

increased training time with more time-steps but stable memory usage. At 15 time-steps, SNNs take

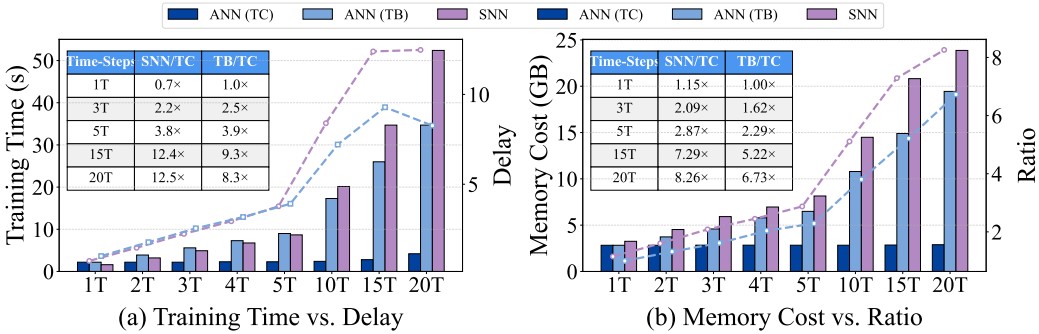

(a) Training Time vs. Delay          (b) Memory Cost vs. Ratio

Figure 5: Speedup and memory cost of using the TC scheme compared to the TB scheme and directly training of SNNs

9× longer than **TC**, while **TC** is 9.3× more efficient—nearly an order of magnitude better in speed and memory than SNN training, validating the ANN-SNN conversion paradigm. Using **TC** to test MT-SNNs on neuromorphic data (Table 6), we achieved 94.10% accuracy with 4.06 time-steps on DVS-Gesture and 84.40% with 2.0 time-steps on CIFAR1-DVS.

## B    COMPARATIVE ANALYSIS OF ERROR GRADIENTS FOR FOUR PARAMETERIZATIONS

Under the Straight-Through Estimator (STE), different parameterizations induce distinct gradient behaviors, leading to varied optimization dynamics. To highlight the superiority of **Combination 4**, we analyze the loss landscape using gradient field visualization, following the methodology in Uhlich et al. (2019).

We sample $10^4$ points from $\mathcal{N}(0,1)$ and define the mean squared error loss $\mathcal{L} = \mathbb{E}[(x - Q(x; \boldsymbol{\eta}))^2]$. Gradients are backpropagated via STE and optimized using SGD across a $50 \times 50$ grid spanning typical parameter ranges. As shown in Figure 3

**Combination 1** ($a$) $[n, s]$: The surface exhibits sharp gradients for large $n$, where $n$ controls the upper bound and $s$ the step size. Their interaction introduces numerous local minima, causing unstable trajectories and slow convergence. Additionally, gradients tend to increase $n$, leading to exponential time-step growth $t = 2^n$, as seen in Figure 4.

**Combination 2** ($b$) $[n, \gamma]$: Both parameters jointly affect the upper bound, resulting in highly irregular gradients. Optimization often prioritizes bit-width alignment before adjusting $\gamma$, increasing the risk of gradient explosion.

**Combination 3** ($c$) $[s, \gamma]$: Without discrete components, the surface is smoother. However, gradients w.r.t. $\gamma$ vanish in non-truncated regions, causing trajectories to slide along the $s$ axis. As $s \to 0$ and $\gamma$ increases, the time-step $t = \gamma/s$ diverges, leading to high latency even under regularization.

---

**Algorithm 1** Conversion Framework for MT-SNNs

---

**Input**: Pre-trained ANN $\{\boldsymbol{W}^l, 2^{n^l}, \gamma^l\}_{l=1}^L$, training data $\mathcal{D}$, epochs $E$, learning rate $\alpha$, penalty $\lambda$
**Output**: Mixed-Timestep SNN $\{\boldsymbol{W}^l, T^l, \theta^l\}_{l=1}^L$

 1: **Phase A: Training of $\boldsymbol{\eta}$**
 2: **for** $l \leftarrow 1$ **to** $L$ **do**
 3:     Replace ReLU by $\mathcal{Q}(\cdot; t^l, \gamma^l)$
 4:     Initialise $t^l \leftarrow 2^{n_{\text{init}}}, \gamma^l \leftarrow \gamma_{\text{init}}$
 5: **end for**
 6: **for** $e \leftarrow 1$ **to** $E$ **do**
 7:     **for each** mini-batch $(\boldsymbol{x}, \boldsymbol{y}) \sim \mathcal{D}$ **do**
 8:         $\boldsymbol{a}^0 \leftarrow \boldsymbol{x}$
 9:         **for** $l \leftarrow 1$ **to** $L$ **do**
10:             $\boldsymbol{z}^l \leftarrow \boldsymbol{W}^l \boldsymbol{a}^{l-1}$
11:             $\hat{\boldsymbol{v}}^l \leftarrow \mathcal{Q}(\boldsymbol{z}^l\, t^l / \gamma^l)$
12:             $\boldsymbol{a}^l \leftarrow \gamma^l\, \hat{\boldsymbol{v}}^l / t^l$
13:         **end for**
14:         $\mathcal{L} \leftarrow \text{CE}(\boldsymbol{a}^L, \boldsymbol{y}) + \lambda \sum_l t^l$
15:         Compute $\nabla_{t^l}\mathcal{L}, \nabla_{\gamma^l}\mathcal{L}$ (STE)
16:         **for** $l \leftarrow 1$ **to** $L$ **do**
17:             $t^l \leftarrow t^l - \alpha(\nabla_{t^l}\mathcal{L} + \lambda)$
18:             $\gamma^l \leftarrow \gamma^l - \alpha\nabla_{\gamma^l}\mathcal{L}$
19:         **end for**
20:     **end for**
21: **end for**
22: **Fix** $T^l \leftarrow t^l, \theta^l \leftarrow \gamma^l$                                    *// model ready for SNN inference*
23: **Phase B: MT-SNNs Forward Inference (Average Time Alignment)**
24: **for** $l \leftarrow 1$ **to** $L$ **do**
25:     $\boldsymbol{q}_t^l \leftarrow \boldsymbol{W}^l \boldsymbol{S}_t^{l-1}\theta^{l-1}$
26:     **if** $l > 1$ **and** $T^{l-1} \neq T^l$ **then**
27:         $\bar{\boldsymbol{q}}^{l-1} \leftarrow \frac{1}{T^{l-1}} \sum_{t=1}^{T^{l-1}} \boldsymbol{q}_t^{l-1}$
28:         $\boldsymbol{q}_{1:T^l}^{l-1} \leftarrow \text{REPEAT}(\bar{\boldsymbol{q}}^{l-1}, T^l)$
29:     **end if**
30:     **for** $t \leftarrow 1$ **to** $T^l$ **do**
31:         $\boldsymbol{U}_t^l \leftarrow \boldsymbol{V}^l + \boldsymbol{q}_t^l$
32:         $\boldsymbol{S}_t^l \leftarrow \mathbb{I}[\boldsymbol{U}_t^l \geq \theta^l]$
33:         $\boldsymbol{V}^l \leftarrow \boldsymbol{U}_t^l - \theta^l \boldsymbol{S}_t^l$                                    *// soft-reset*
34:     **end for**
35: **end for**

---

**Combination 4** (*d*) $[t, \gamma]$: This configuration yields the smoothest landscape, stable gradients, and fastest convergence. It avoids the instability of discrete parameters while enabling direct control over latency via $t$, making it the most effective formulation.

## C  ENERGY EFFICIENCY ANALYSIS.

In ANN, the output activations are obtained by performing a multiplication and accumulation operation between the input activations and the weights. This operation is referred to as *multiply-accumulate operation (MACs)*. For example, $x^l = W^l \cdot x^{l-1}$, where $x^{l-1}$ is the output activation from the previous layer, and $W^l$ is the weight matrix of the current layer. The elements of $x^{l-1}$ and $W^l$ are typically multi-bit fixed-point or floating-point values. In SNN, the activations are represented by binary spikes, meaning the activation values are either 0 or 1. As a result, the MACs in SNN can be simplified to an *accumulation operation (ACs)* over the weights . For example, given an activation $x^{l-1} = [0, 1, 0, 1]^T$ and weights $W^l = [w_1, w_2, w_3, w_4]$, the output is computed as $x^l = W^l \cdot x^{l-1} = w_2 + w_4$. Addition operations typically consume less energy than multiplication

operations in hardware. This is because multiplication involves more complex computational logic and requires more hardware resources (e.g., multipliers), while addition only requires simple adder units. The specific energy consumption comparison is shown in Table 7.

To compare the energy efficiency of SNN and quantized ANN, we conducted experiments using VGG-16 and ResNet-34 on ImageNet-1k. We assume that the weights are all 8-bit fixed-point quantized (this assumption does not affect the final comparison, as both SNN and ANN use the same weights). The energy consumption of multiplication and addition operations for both fixed-point and floating-point (32-bit and 8-bit) is provided in (Horowitz, 2014). Using this data, we can calculate the energy required for MACs. For example, for 8-bit fixed-point quantized activations and weights, the energy consumption of a multiply-accumulate operation is 0.23 pJ.

Table 7: Comparison of Energy Efficiency Between ANN and SNN

| Model | Energy (mJ) | | $E_{ANN}$ / $E_{SNN}$ |
| --- | --- | --- | --- |
| | ANN | SNN | |
| **CIFAR-10** | | | |
| VGG-16 | 0.0233 | 0.0041 | 5.68× |
| ResNet-18 | 0.0411 | 0.0073 | 5.63× |
| ResNet-20 | 0.0034 | 0.0011 | 3.09× |
| **CIFAR-100** | | | |
| VGG-16 | 0.0242 | 0.0049 | 4.94× |
| ResNet-18 | 0.0438 | 0.0089 | 4.92× |
| ResNet-20 | 0.0038 | 0.0016 | 2.38× |

Table 8: Energy Table for 45nm CMOS Process

| Operation | Energy (pJ) |
| --- | --- |
| 8b ADD (INT) | 0.03 |
| 32b ADD (INT) | 0.1 |
| 8b MULT (INT) | 0.2 |
| 32b MULT (INT) | 3.1 |
| 8b MAC (INT) | 0.23 |
| 16b ADD (FP) | 0.4 |
| 32b ADD (FP) | 0.9 |
| 16b MULT (FP) | 1.1 |
| 32b MULT (FP) | 3.7 |
| 1b-8b MULT (INT)[*] | 0.025 |
| 2b-8b MULT (INT)[*] | 0.05 |
| 3b-8b MULT (INT)[*] | 0.075 |

[*] Data derived from Nagendra et al. (1996) and Potipireddi & Asati (2013)

However, because the activations of ANN in our work MT-SNNs are mixed-precision quantized, the bit width of activations varies across different layers, which results in multiply-accumulate operations (MACs) with different bit-width activations and weights, such as INT2 activations and INT8 weights, for which energy consumption values are not provided in (Horowitz, 2014). To address this, we refer to the approach in (Nagendra et al., 1996), which states that the power consumption of an adder is linearly related to bitwidth, while the power consumption of a multiplier is quadratic with respect to bitwidth (Potipireddi & Asati, 2013). This allows us to compute the energy of multiply-accumulate operations for different bitwidths. For instance, a 2-bit fixed-point multiplication (INT2) consumes 1/16 of the energy of an 8-bit fixed-point multiplication (INT8). Since SNN use binary spikes for activations, only accumulation operations (ACs) are required. Therefore, we can directly utilize the data from (Horowitz, 2014) without needing additional energy data from (Nagendra et al., 1996; Potipireddi & Asati, 2013).

We follow the energy calculation method from (Panda et al., 2020), with $N$ input channels, $M$ output channels, weight kernel size $k \times k$, and output size $O \times O$, the total FLOPS for ANN and SNN are as described in the equations below.

$$FLOPS_{ANN} = O^2 \times N \times k^2 \times M, \quad FLOPS_{SNN} = O^2 \times N \times k^2 \times M \times f_r \quad (22)$$

Where $f_r$ represents the total number of firing rate per layer, $f_r \ll 1$ in SNNs. For energy calculation, MAC (for ANN) and AC (Addition operation, for SNN) as below:

$$E_{ANN} = E_{MAC}^l \times \sum_{l=1}^{N} FLOPS_{ANN}, \quad E_{SNN} = E_{AC} \times T_l \times \sum_{l=1}^{N} FLOPS_{SNN} \quad (23)$$

Where $E_{MAC}^l$ represents the energy per MAC operation in l-th layer, the $E_{MAC}^l$ vary across layers due to the different quantization bit-widths of activations in each layer, $E_{AC}$ represents the energy consumed by addition operations, which is 0.03 pJ for INT8 quantized weights, and the $E_{AC}$ value is the same across all layers. $T_l$ denotes the time-step of the $l$-th layer in the SNN, and the time-steps may vary across different layers.

## D  TEMPORAL SCALABILITY ANALYSIS

To evaluate the proposed MT-SNNs's temporal scalability, we investigate how its accuracy evolves with increasing timesteps, focusing on two goals: (1) verifying whether performance improves or degrades across a broader temporal scale, and (2) confirming the optimality of its time-step configuration. Experiments use ResNet-18, ResNet-20, VGG-16 on CIFAR-10/CIFAR-100. We incrementally double each layer's timesteps (via scaling) and monitor accuracy changes. As shown in Figure 6, MT-SNNs accuracy rises gradually with more timesteps—even outperforming baseline quantized ANNs at moderate scaling. Beyond four times the original timesteps, accuracy saturates with no further gains. This confirms effective temporal scalability within a reasonable range and validates the initial time-step design's optimality.

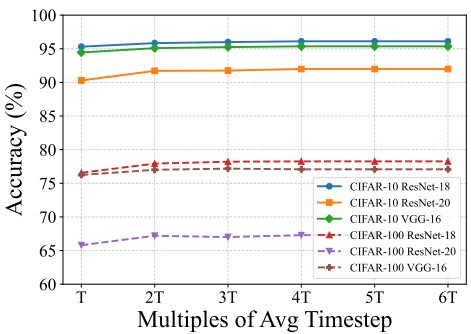

Figure 6: Temporal Scalability Analysis.

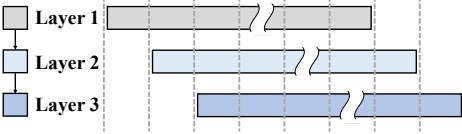

Figure 7: Pipeline Analysis.

---

**Algorithm 2** Neuron and Temporal Loops

**Input**: Input spikes $I_{\text{spikes}}$; Weights $W$; Dimensions $C_o, M, H_o, W_o, N, T, S, K_h, K_w, C_i, V$

**Output**: $V_{\text{Next}}$ and $O_{\text{spikes}}$ after processing

1:  **for** $o \leftarrow 0$ **to** $C_o/M$ **do**
2:      **for** $h \leftarrow 0$ **to** $H_o$ **do**
3:          **for** $w \leftarrow 0$ **to** $W_o/N$ **do**
4:              **for** $t \leftarrow 0$ **to** $T/S$ **do**
5:                  **for** $k_h \leftarrow 0$ **to** $K_h$ **do**
6:                      **for** $k_w \leftarrow 0$ **to** $K_w$ **do**
7:                          **for** $i \leftarrow 0$ **to** $C_i/V$ **do**
8:                              $P_{\text{sum}} += W \times I_{\text{spikes}}$
9:                          **end for**
10:                     **end for**
11:                 **end for**
12:                 $V_{\text{Next}}, O_{\text{spikes}} \leftarrow \text{Node}(P_{\text{sum}}, V_{\text{Pre}})$
13:             **end for**
14:         **end for**
15:     **end for**
16: **end for**

---

## E  HARDWARE DEPLOYMENT EFFICIENCY ANALYSIS

Mainstream SNN hardware implementations fall into two categories: ANN accelerator variants and Non-Von Neumann distributed multi-core architectures (e.g., TrueNorth Akopyan et al. (2015), Loihi Davies et al. (2018)).

ANN accelerator variants achieve asynchronous computation primarily by routing non-zero inputs to processing element (PE) arrays for spike-based matrix calculations. They compute only a subset of the neural network at a time Li et al. (2024), iterating to cover the full network; Algorithm 2 details this data flow, with each layer using a unified time-step $T$. For MT-SNNs, layers have distinct timesteps, and average temporal alignment only averages each layer's output along the temporal dimension—this preserves the original data flow, enabling MT-SNNs deployment on such hardware.

In contrast, multi-core neuromorphic hardware distributes neurons of all layers across separate cores: upon receiving spike events, neurons immediately perform spike-based computations for asynchronous execution, and the network runs in a pipeline (Figure 7): at $T_1$, Layer 1 processes Sample 1; at $T_2$, Layer 2 processes Layer 1's output (Sample 1) while Layer 1 processes Sample 2. MT-SNNs require temporal alignment to handle layer-specific timesteps—Layer 2 must wait for Layer 1 to complete $T_{l1}$ timesteps before starting. Though MT-SNNs run on this hardware, pipeline stalling may occur, introducing delays and hindering optimal performance.

# F  GRADIENT DERIVATION FOR FOUR PARAMETER COMBINATIONS

For the regularized mixed time-step optimization problem, the goal is to minimize the cost function $\mathcal{J}(\boldsymbol{\eta^l})$, which combines a quantization loss term and a time-step regularization term, formulated as:

$$\min_{\boldsymbol{\eta^l}} \mathcal{J}(\boldsymbol{\eta^l}) = \mathcal{L}\left(\mathcal{Q}\left(\boldsymbol{a^l}; \boldsymbol{\eta^l}\right)\right) + \lambda \sum_{l=1}^{L} t^l \tag{24}$$

where $\mathcal{L}(\cdot)$ is the loss for quantized output $\boldsymbol{a^l}$ (from quantizer $\mathcal{Q}(\cdot)$ with parameters $\boldsymbol{\eta^l}$), $\lambda$ is the regularization coefficient, and $t^l$ is the $l$-th time-step.

To optimize $\boldsymbol{\eta^l}$, the gradient of $\mathcal{J}$ with respect to $\boldsymbol{\eta^l}$ decomposing loss and regularization contributions is:

$$\frac{\partial \mathcal{J}}{\partial \boldsymbol{\eta^l}} = \frac{\partial \mathcal{L}}{\partial \boldsymbol{a^l}} \cdot \frac{\partial \boldsymbol{a^l}}{\partial \boldsymbol{\eta^l}} + \lambda \frac{\partial t^l}{\partial \boldsymbol{\eta^l}} \tag{25}$$

Here, $\frac{\partial \mathcal{L}}{\partial \boldsymbol{a^l}}$ is the loss to $\boldsymbol{a^l}$, while $\frac{\partial \boldsymbol{a^l}}{\partial \boldsymbol{\eta^l}}$ and $\frac{\partial t^l}{\partial \boldsymbol{\eta^l}}$ describe how $\boldsymbol{a^l}$ and $t^l$ vary with $\boldsymbol{\eta^l}$.

The basic quantization formula maps input $\boldsymbol{z}^l$ to $\boldsymbol{a^l}$ via scaling (by $\gamma^l$, $n^l$) and clipping , given by:

$$\boldsymbol{a^l} = \frac{\gamma^l}{2^{n^l}} \cdot \text{clip}\left(\mathcal{Q}\left(\boldsymbol{z}^l \cdot \frac{2^{n^l}}{\gamma^l}\right); 0, 2^{n^l}\right) \tag{26}$$

where $\text{clip}(\cdot; 0, 2^{n^l})$ restricts values to $[0, 2^{n^l}]$, and $\frac{\gamma^l}{2^{n^l}}$ scales back to the original domain.

A binary mask function $\boldsymbol{I}(C)$ handles conditional operations in later derivations, defined as:

$$\boldsymbol{I}(C) = \begin{cases} 1 & \text{if condition } C \text{ is true} \\ 0 & \text{otherwise} \end{cases} \tag{27}$$

The equivalent time-step $t^l$ varies by parameter combination, with the piecewise definition:

$$t^l = \begin{cases} 2^{n^l} & \text{Combinations 1, 2} \\ \frac{\gamma^l}{s^l} & \text{Combination 3} \\ t^l & \text{Combination 4} \end{cases} \tag{28}$$

## F.1  COMBINATION 1

The optimization parameter vector is $\boldsymbol{\eta^l} = [n^l, s^l]^T$, and the quantization step satisfies $\gamma^l = s^l \cdot 2^{n^l}$ (linking $\gamma^l$ to adjustable parameters). Substitute the constraint into the basic formula:

$$\boldsymbol{a^l} = s^l \cdot \text{clip}\left(\mathcal{Q}\left(\frac{\boldsymbol{z}^l}{s^l}\right); 0, 2^{n^l}\right) \tag{29}$$

To decompose the clipping operation, define $\hat{z}^l = \mathcal{Q}(\boldsymbol{z}^l/s^l)$ and use the binary mask $\boldsymbol{I}(C)$ (introduced earlier). This rewrites the clipped term as a weighted sum of valid/overflow values, giving:

$$\boldsymbol{a^l} = s^l \cdot \left[\hat{z}^l \cdot \boldsymbol{I}(\hat{z}^l \leq 2^{n^l}) + 2^{n^l} \cdot \boldsymbol{I}(\hat{z}^l > 2^{n^l})\right] \tag{30}$$

Derive $\partial \boldsymbol{a^l}/\partial n^l$, since $\hat{z}^l$ is independent of $n^l$, only $2^{n^l}$ contributes, using $\partial 2^{n^l}/\partial n^l = 2^{n^l} \ln(2)$:

$$\frac{\partial \boldsymbol{a^l}}{\partial n^l} = s^l \cdot \frac{\partial}{\partial n^l}\left[2^{n^l} \cdot \boldsymbol{I}(\hat{z}^l > 2^{n^l})\right] = s^l \cdot 2^{n^l} \ln(2) \cdot \boldsymbol{I}(\hat{z}^l > 2^{n^l}) \tag{31}$$

Compute $\partial \boldsymbol{a^l}/\partial s^l$ via product rule (for $s^l \cdot \hat{z}^l$) and accounting for $\hat{z}^l$'s dependence on $s^l$:

$$\frac{\partial \boldsymbol{a^l}}{\partial s^l} = \hat{z}^l \cdot \boldsymbol{I}(\hat{z}^l \leq 2^{n^l}) + 2^{n^l} \cdot \boldsymbol{I}(\hat{z}^l > 2^{n^l}) + s^l \cdot \frac{\partial \hat{z}^l}{\partial s^l} \cdot \boldsymbol{I}(\hat{z}^l \leq 2^{n^l}) \tag{32}$$

Under Straight-Through Estimator (STE) assumption ($\partial \mathcal{Q}(\cdot)/\partial s^l \approx 1$), $\partial \hat{z}^l/\partial s^l = -\boldsymbol{z}^l/(s^l)^2$. Substitute and simplify with element-wise multiplication ($\odot$):

$$\frac{\partial \boldsymbol{a^l}}{\partial s^l} = (\hat{z}^l - \frac{\boldsymbol{z}^l}{s^l}) \odot \boldsymbol{I}(\hat{z}^l \leq 2^{n^l}) + 2^{n^l} \cdot \boldsymbol{I}(\hat{z}^l > 2^{n^l}) \tag{33}$$

For the regularization term, $t^l = 2^{n^l}$. Its derivatives are:

$$\frac{\partial t^l}{\partial n^l} = 2^{n^l} \ln(2) \tag{34}$$

$$\frac{\partial t^l}{\partial s^l} = 0 \tag{35}$$

Substitute $\partial \boldsymbol{a^l}/\partial n^l$ and $\partial t^l/\partial n^l$ into the total loss gradient, factoring out $2^{n^l} \ln(2)$:

$$\frac{\partial \mathcal{J}}{\partial n^l} = \frac{\partial \mathcal{L}}{\partial \boldsymbol{a^l}} \cdot s^l \cdot 2^{n^l} \ln(2) \cdot \boldsymbol{I}(\hat{\boldsymbol{z}}^l > 2^{n^l}) + \lambda \cdot 2^{n^l} \ln(2)$$

$$= \left[ \frac{\partial \mathcal{L}}{\partial \boldsymbol{a^l}} \cdot s^l \odot \boldsymbol{I}(\hat{\boldsymbol{z}}^l > 2^{n^l}) + \lambda \right] \cdot 2^{n^l} \ln(2) \tag{36}$$

Finally, substitute $\partial \boldsymbol{a^l}/\partial s^l$ (and $\partial t^l/\partial s^l = 0$) into the total loss gradient to get $\partial \mathcal{J}/\partial s^l$:

$$\frac{\partial \mathcal{J}}{\partial s^l} = \frac{\partial \mathcal{L}}{\partial \boldsymbol{a^l}} \cdot \left[ (\hat{\boldsymbol{z}}^l - \frac{\boldsymbol{z}^l}{s^l}) \odot \boldsymbol{I}(\hat{\boldsymbol{z}}^l \le 2^{n^l}) + 2^{n^l} \cdot \boldsymbol{I}(\hat{\boldsymbol{z}}^l > 2^{n^l}) \right] \tag{37}$$

## F.2 COMBINATION 2

The optimization parameter vector is $\boldsymbol{\eta^l} = [n^l, \gamma^l]^T$, with the scaling factor $s^l$ constrained by $s^l = \gamma^l/2^{n^l}$. To simplify the original quantization formula, we define an intermediate quantized variable $\hat{\boldsymbol{u}}^l = \mathcal{Q}(\boldsymbol{z}^l \cdot 2^{n^l}/\gamma^l)$, which captures the quantized result of $\boldsymbol{z}^l$ scaled by $2^{n^l}/\gamma^l$. Substituting $\hat{\boldsymbol{u}}^l$ and using the binary mask function $\boldsymbol{I}(C)$ to decompose the clipping operation:

$$\boldsymbol{a}^l = \frac{\gamma^l}{2^{n^l}} \cdot \left[ \hat{\boldsymbol{u}}^l \cdot \boldsymbol{I}(\hat{\boldsymbol{u}}^l \le 2^{n^l}) + 2^{n^l} \cdot \boldsymbol{I}(\hat{\boldsymbol{u}}^l > 2^{n^l}) \right] \tag{38}$$

Next, we compute the partial derivative of $\boldsymbol{a}^l$ with respect to $n^l$. Since $\boldsymbol{a}^l$ is a product of two terms ($\gamma^l/2^{n^l}$ and the clipped sum), we apply the product rule for differentiation. The derivative expression before simplification is:

$$\frac{\partial \boldsymbol{a}^l}{\partial n^l} = \frac{\partial}{\partial n^l} \left[ \frac{\gamma^l}{2^{n^l}} \right] \cdot [\hat{\boldsymbol{u}}^l \cdot \boldsymbol{I}(\hat{\boldsymbol{u}}^l \le 2^{n^l}) + 2^{n^l} \cdot \boldsymbol{I}(\hat{\boldsymbol{u}}^l > 2^{n^l})]$$

$$+ \frac{\gamma^l}{2^{n^l}} \cdot \frac{\partial}{\partial n^l} [\hat{\boldsymbol{u}}^l \cdot \boldsymbol{I}(\hat{\boldsymbol{u}}^l \le 2^{n^l}) + 2^{n^l} \cdot \boldsymbol{I}(\hat{\boldsymbol{u}}^l > 2^{n^l})] \tag{39}$$

where, $\frac{\partial}{\partial n^l}[\frac{\gamma^l}{2^{n^l}}] = -\frac{\gamma^l \ln(2)}{2^{n^l}}$, $\frac{\partial \hat{\boldsymbol{u}}^l}{\partial n^l} = \frac{\boldsymbol{z}^l \cdot 2^{n^l} \ln(2)}{\gamma^l}$, $\frac{\partial}{\partial n^l}[2^{n^l}] = 2^{n^l} \ln(2)$. Substituting the three key derivatives and simplifying, we obtain the final form of $\partial \boldsymbol{a}^l/\partial n^l$:

$$\frac{\partial \boldsymbol{a}^l}{\partial n^l} = (\boldsymbol{z}^l - s^l \cdot \hat{\boldsymbol{u}}^l) \ln(2) \odot \boldsymbol{I}(\hat{\boldsymbol{u}}^l \le 2^{n^l}) + \gamma^l \ln(2) \cdot \boldsymbol{I}(\hat{\boldsymbol{u}}^l > 2^{n^l}) \tag{40}$$

We then derive the partial derivative of $\boldsymbol{a}^l$ with respect to $\gamma^l$. Under the STE assumption, the derivative of $\hat{\boldsymbol{u}}^l$ with respect to $\gamma^l$ is $\partial \hat{\boldsymbol{u}}^l/\partial \gamma^l = -\boldsymbol{z}^l \cdot 2^{n^l}/(\gamma^l)^2$. The derivative expression is:

$$\frac{\partial \boldsymbol{a}^l}{\partial \gamma^l} = \frac{1}{2^{n^l}} \cdot [\hat{\boldsymbol{u}}^l \cdot \boldsymbol{I}(\hat{\boldsymbol{u}}^l \le 2^{n^l}) + 2^{n^l} \cdot \boldsymbol{I}(\hat{\boldsymbol{u}}^l > 2^{n^l})] + \frac{\gamma^l}{2^{n^l}} \cdot \frac{\partial \hat{\boldsymbol{u}}^l}{\partial \gamma^l} \cdot \boldsymbol{I}(\hat{\boldsymbol{u}}^l \le 2^{n^l})$$

Substituting $\partial \hat{\boldsymbol{u}}^l/\partial \gamma^l$ and simplifying, the derivative simplifies to:

$$\frac{\partial \boldsymbol{a}^l}{\partial \gamma^l} = (s^l \cdot \hat{\boldsymbol{u}}^l - \frac{\boldsymbol{z}^l}{\gamma^l}) \odot \boldsymbol{I}(\hat{\boldsymbol{u}}^l \le 2^{n^l}) + \boldsymbol{I}(\hat{\boldsymbol{u}}^l > 2^{n^l}) \tag{41}$$

For the regularization term, recall from prior sections that $t^l = 2^{n^l}$ for this combination. Thus, $t^l$ depends only on $n^l$, leading to the following derivatives:

$$\frac{\partial t^l}{\partial n^l} = 2^{n^l} \ln(2) \tag{42}$$

$$\frac{\partial t^l}{\partial \gamma^l} = 0 \tag{43}$$

Using the total loss function gradient formula , we substitute $\partial \boldsymbol{a}^l / \partial n^l$ and $\partial t^l / \partial n^l$ to compute the gradient of $\mathcal{J}$ with respect to $n^l$:

$$\frac{\partial \mathcal{J}}{\partial n^l} = \frac{\partial \mathcal{L}}{\partial \boldsymbol{a}^l} \cdot (\boldsymbol{z}^l - s^l \cdot \hat{\boldsymbol{u}}^l) \ln(2) \odot \boldsymbol{I}(\hat{\boldsymbol{u}}^l \leq 2^{n^l}) + \left[ \frac{\partial \mathcal{L}}{\partial \boldsymbol{a}^l} \cdot \gamma^l \odot \boldsymbol{I}(\hat{\boldsymbol{u}}^l > 2^{n^l}) + \lambda \right] \ln(2) \quad (44)$$

Finally, substituting $\partial \boldsymbol{a}^l / \partial \gamma^l$ and $\partial t^l / \partial \gamma^l = 0$ into the total loss gradient formula gives the gradient of $\mathcal{J}$ with respect to $\gamma^l$:

$$\frac{\partial \mathcal{J}}{\partial \gamma^l} = \frac{\partial \mathcal{L}}{\partial \boldsymbol{a}^l} \cdot \left[ (s^l \cdot \hat{\boldsymbol{u}}^l - \frac{\boldsymbol{z}^l}{\gamma^l}) \odot \boldsymbol{I}(\hat{\boldsymbol{u}}^l \leq 2^{n^l}) + \boldsymbol{I}(\hat{\boldsymbol{u}}^l > 2^{n^l}) \right] \quad (45)$$

### F.3 COMBINATION 3

The optimization parameter vector is defined as $\boldsymbol{\eta^l} = [s^l, \gamma^l]^T$, with the bit-width-related term constrained by $2^{n^l} = \gamma^l / s^l$ . To simplify the basic quantization formula, we first substitute this constraint into the expression for $\boldsymbol{a}^l$. We also retain the intermediate quantized variable $\hat{\boldsymbol{z}}^l = \mathcal{Q}(\boldsymbol{z}^l / s^l)$ to represent the quantized result of $\boldsymbol{z}^l$ scaled by $1/s^l$. Using the binary mask function $\boldsymbol{I}(C)$ to handle clipping, the substituted quantization formula becomes:

$$\boldsymbol{a}^l = s^l \cdot \left[ \hat{\boldsymbol{z}}^l \cdot \boldsymbol{I}(\hat{\boldsymbol{z}}^l \leq \frac{\gamma^l}{s^l}) + \frac{\gamma^l}{s^l} \cdot \boldsymbol{I}(\hat{\boldsymbol{z}}^l > \frac{\gamma^l}{s^l}) \right] \quad (46)$$

Next, we compute the partial derivative of $\boldsymbol{a}^l$ with respect to $s^l$. Due to the product structure of $\boldsymbol{a}^l$ and the dependence of the clipped threshold $(\gamma^l / s^l)$ on $s^l$, the derivative requires applying the product rule and accounting for the threshold's variation. The derivative expression (before substituting key terms) includes three contributions: the direct derivative of $s^l$, the derivative of $\hat{\boldsymbol{z}}^l$ with respect to $s^l$, and the derivative of the threshold $\gamma^l / s^l$ with respect to $s^l$:

$$\frac{\partial \boldsymbol{a}^l}{\partial s^l} = \hat{\boldsymbol{z}}^l \cdot \boldsymbol{I}(\hat{\boldsymbol{z}}^l \leq \frac{\gamma^l}{s^l}) + \frac{\gamma^l}{s^l} \cdot \boldsymbol{I}(\hat{\boldsymbol{z}}^l > \frac{\gamma^l}{s^l})$$
$$+ s^l \cdot \frac{\partial \hat{\boldsymbol{z}}^l}{\partial s^l} \cdot \boldsymbol{I}(\hat{\boldsymbol{z}}^l \leq \frac{\gamma^l}{s^l})$$
$$+ s^l \cdot \frac{\partial}{\partial s^l}[\frac{\gamma^l}{s^l}] \cdot \boldsymbol{I}(\hat{\boldsymbol{z}}^l > \frac{\gamma^l}{s^l}) \quad (47)$$

Two key derivative results are used here $\frac{\partial \hat{\boldsymbol{z}}^l}{\partial s^l} = -\frac{\boldsymbol{z}^l}{(s^l)^2}, \frac{\partial}{\partial s^l}[\frac{\gamma^l}{s^l}] = -\frac{\gamma^l}{(s^l)^2}$ derived under the Straight-Through Estimator, STE, assumption). Substituting these two derivatives into the expression and simplifying (noting that the second and fourth terms cancel out) yields the final form of $\partial \boldsymbol{a}^l / \partial s^l$:

$$\frac{\partial \boldsymbol{a}^l}{\partial s^l} = \hat{\boldsymbol{z}}^l \cdot \boldsymbol{I}(\hat{\boldsymbol{z}}^l \leq \frac{\gamma^l}{s^l}) + \frac{\gamma^l}{s^l} \cdot \boldsymbol{I}(\hat{\boldsymbol{z}}^l > \frac{\gamma^l}{s^l})$$
$$- \frac{\boldsymbol{z}^l}{s^l} \cdot \boldsymbol{I}(\hat{\boldsymbol{z}}^l \leq \frac{\gamma^l}{s^l}) - \frac{\gamma^l}{s^l} \cdot \boldsymbol{I}(\hat{\boldsymbol{z}}^l > \frac{\gamma^l}{s^l})$$
$$= (\hat{\boldsymbol{z}}^l - \frac{\boldsymbol{z}^l}{s^l}) \odot \boldsymbol{I}(\hat{\boldsymbol{z}}^l \leq \frac{\gamma^l}{s^l}) \quad (48)$$

We then derive the partial derivative of $\boldsymbol{a}^l$ with respect to $\gamma^l$. Only the clipping threshold term $(\gamma^l / s^l)$ depends on $\gamma^l$, so the derivative simplifies to the product of $s^l$ (from the outer scaling) and the derivative of $\gamma^l / s^l$ with respect to $\gamma^l$, activated by the mask for overflow values. This results in:

$$\frac{\partial \boldsymbol{a}^l}{\partial \gamma^l} = s^l \cdot \frac{\partial}{\partial \gamma^l}[\frac{\gamma^l}{s^l}] \cdot \boldsymbol{I}(\hat{\boldsymbol{z}}^l > \frac{\gamma^l}{s^l}) = \boldsymbol{I}(\hat{\boldsymbol{z}}^l > \frac{\gamma^l}{s^l}) \quad (49)$$

For the regularization term, recall that $t^l = \gamma^l / s^l$ for Combination 3. Thus, $t^l$ depends on both $s^l$ and $\gamma^l$, leading to the following derivatives:

$$\frac{\partial t^l}{\partial s^l} = -\frac{\gamma^l}{(s^l)^2} \quad (50)$$

$$\frac{\partial t^l}{\partial \gamma^l} = \frac{1}{s^l} \quad (51)$$

Using the total loss function gradient formula, we substitute $\partial \boldsymbol{a}^l / \partial s^l$ and $\partial t^l / \partial s^l$ to compute the gradient of $\mathcal{J}$ with respect to $s^l$:

$$\frac{\partial \mathcal{J}}{\partial s^l} = \frac{\partial \mathcal{L}}{\partial \boldsymbol{a^l}} \cdot (\hat{\boldsymbol{z}}^l - \frac{\boldsymbol{z}^l}{s^l}) \odot \boldsymbol{I}(\hat{\boldsymbol{z}}^l \leq \frac{\gamma^l}{s^l}) - \lambda \frac{\gamma^l}{(s^l)^2} \tag{52}$$

Finally, substituting $\partial \boldsymbol{a}^l / \partial \gamma^l$ and $\partial t^l / \partial \gamma^l$ into the total loss gradient formula gives the gradient of $\mathcal{J}$ with respect to $\gamma^l$:

$$\frac{\partial \mathcal{J}}{\partial \gamma^l} = \frac{\partial \mathcal{L}}{\partial \boldsymbol{a^l}} \cdot \boldsymbol{I}(\hat{\boldsymbol{z}}^l > \frac{\gamma^l}{s^l}) + \frac{\lambda}{s^l} \tag{53}$$

### F.4 COMBINATION 4

The optimization parameter vector is $\boldsymbol{\eta^l} = [t^l, \gamma^l]^T$, with the scaling factor constrained by $s^l = \gamma^l / t^l$. To simplify the basic quantization formula, we first substitute this constraint and define an intermediate quantized variable $\hat{\boldsymbol{v}}^l = \mathcal{Q}(\boldsymbol{z}^l \cdot t^l / \gamma^l)$, the variable represents the quantized result of $\boldsymbol{z}^l$ scaled by $t^l / \gamma^l$. Using the binary mask function $\boldsymbol{I}(C)$ to decompose the clipping operation, the substituted quantization formula becomes:

$$\boldsymbol{a}^l = \frac{\gamma^l}{t^l} \cdot \left[ \hat{\boldsymbol{v}}^l \cdot \boldsymbol{I}(\hat{\boldsymbol{v}}^l \leq t^l) + t^l \cdot \boldsymbol{I}(\hat{\boldsymbol{v}}^l > t^l) \right] \tag{54}$$

Next, we compute the partial derivative of $\boldsymbol{a}^l$ with respect to $t^l$. Since $\boldsymbol{a}^l$ is a product of $\gamma^l / t^l$ and the clipped sum, we apply the product rule. Three key derivative results are used here, $\frac{\partial}{\partial t^l}[\frac{\gamma^l}{t^l}] = -\frac{\gamma^l}{(t^l)^2}$, $\frac{\partial \hat{\boldsymbol{v}}^l}{\partial t^l} = \frac{\boldsymbol{z}^l}{\gamma^l}$, $\frac{\partial}{\partial t^l}[t^l] = 1$. The derivative expression before simplification is:

$$\frac{\partial \boldsymbol{a}^l}{\partial t^l} = \frac{\partial}{\partial t^l}[\frac{\gamma^l}{t^l}] \cdot [\hat{\boldsymbol{v}}^l \cdot \boldsymbol{I}(\hat{\boldsymbol{v}}^l \leq t^l) + t^l \cdot \boldsymbol{I}(\hat{\boldsymbol{v}}^l > t^l)]$$

$$+ \frac{\gamma^l}{t^l} \cdot \frac{\partial}{\partial t^l}[\hat{\boldsymbol{v}}^l \cdot \boldsymbol{I}(\hat{\boldsymbol{v}}^l \leq t^l) + t^l \cdot \boldsymbol{I}(\hat{\boldsymbol{v}}^l > t^l)] \tag{55}$$

Substituting the three key derivatives and simplifying yields the final form of $\partial \boldsymbol{a}^l / \partial t^l$:

$$\frac{\partial \boldsymbol{a}^l}{\partial t^l} = -\frac{\gamma^l}{(t^l)^2} \cdot \hat{\boldsymbol{v}}^l \cdot \boldsymbol{I}(\hat{\boldsymbol{v}}^l \leq t^l) + \frac{\boldsymbol{z}^l}{t^l} \cdot \boldsymbol{I}(\hat{\boldsymbol{v}}^l \leq t^l)$$

$$+ \frac{\gamma^l}{t^l} \cdot \boldsymbol{I}(\hat{\boldsymbol{v}}^l > t^l)$$

$$= (\frac{\boldsymbol{z}^l}{t^l} - \frac{\gamma^l \hat{\boldsymbol{v}}^l}{(t^l)^2}) \odot \boldsymbol{I}(\hat{\boldsymbol{v}}^l \leq t^l) + \frac{\gamma^l}{t^l} \cdot \boldsymbol{I}(\hat{\boldsymbol{v}}^l > t^l) \tag{56}$$

We then derive the partial derivative of $\boldsymbol{a}^l$ with respect to $\gamma^l$, again applying the product rule. Under the STE assumption, the derivative of $\hat{\boldsymbol{v}}^l$ with respect to $\gamma^l$ is $\partial \hat{\boldsymbol{v}}^l / \partial \gamma^l = -\boldsymbol{z}^l \cdot t^l / (\gamma^l)^2$. The initial derivative expression is:

$$\frac{\partial \boldsymbol{a}^l}{\partial \gamma^l} = \frac{1}{t^l} \cdot [\hat{\boldsymbol{v}}^l \cdot \boldsymbol{I}(\hat{\boldsymbol{v}}^l \leq t^l) + t^l \cdot \boldsymbol{I}(\hat{\boldsymbol{v}}^l > t^l)] + \frac{\gamma^l}{t^l} \cdot \frac{\partial \hat{\boldsymbol{v}}^l}{\partial \gamma^l} \cdot \boldsymbol{I}(\hat{\boldsymbol{v}}^l \leq t^l) \tag{57}$$

Substituting $\partial \hat{\boldsymbol{v}}^l / \partial \gamma^l$ and simplifying (using $s^l = \gamma^l / t^l$ implicitly for term grouping) gives:

$$\frac{\partial \boldsymbol{a}^l}{\partial \gamma^l} = (\frac{\hat{\boldsymbol{v}}^l}{t^l} - \frac{\boldsymbol{z}^l}{\gamma^l}) \odot \boldsymbol{I}(\hat{\boldsymbol{v}}^l \leq t^l) + \boldsymbol{I}(\hat{\boldsymbol{v}}^l > t^l) \tag{58}$$

For regularization term, $\frac{\partial t^l}{\partial t^l} = 1$, $\frac{\partial t^l}{\partial \gamma^l} = 0$. Using the total loss function gradient formula, we substitute $\partial \boldsymbol{a}^l / \partial t^l$ and $\partial t^l / \partial t^l = 1$ to compute the gradient of $\mathcal{J}$ with respect to $t^l$:

$$\frac{\partial \mathcal{J}}{\partial t^l} = \frac{\partial \mathcal{L}}{\partial \boldsymbol{a^l}} \cdot \left[ (\frac{\boldsymbol{z}^l}{t^l} - \frac{\gamma^l \hat{\boldsymbol{v}}^l}{(t^l)^2}) \odot \boldsymbol{I}(\hat{\boldsymbol{v}}^l \leq t^l) + \frac{\gamma^l}{t^l} \cdot \boldsymbol{I}(\hat{\boldsymbol{v}}^l > t^l) \right] + \lambda \tag{59}$$

Finally, substituting $\partial \boldsymbol{a}^l / \partial \gamma^l$ and $\partial t^l / \partial \gamma^l = 0$ into the total loss gradient formula yields the gradient of $\mathcal{J}$ with respect to $\gamma^l$:

$$\frac{\partial \mathcal{J}}{\partial \gamma^l} = \frac{\partial \mathcal{L}}{\partial \boldsymbol{a^l}} \cdot \left[ (\frac{\hat{\boldsymbol{v}}^l}{t^l} - \frac{\boldsymbol{z}^l}{\gamma^l}) \odot \boldsymbol{I}(\hat{\boldsymbol{v}}^l \leq t^l) + \boldsymbol{I}(\hat{\boldsymbol{v}}^l > t^l) \right] \tag{60}$$

## G  THE USE OF LARGE LANGUAGE MODELS (LLMS)

LLMs are used to assist in checking grammatical and spelling errors, polishing the language expression logic of papers, aiding in the derivation of Formula F, and supporting the creation of paper figures.

