# OpenReview forum: "Mixed-Timestep Spiking Neural Networks with Temporal Alignment for Ultra-Low Latency Conversion"
_ICLR.cc/2026/Conference — ICLR 2026 Conference Withdrawn Submission_

### Official Review · Reviewer_4CD7 · 2025-10-26

**Soundness:** 1
**Presentation:** 2
**Contribution:** 2
**Rating:** 2
**Confidence:** 5

**Summary:**

The authors proposed a Mixed-Timestep Spiking Neural Network to allow each layer to operate in different timestep settings, thereby saving latency.

**Strengths:**

The presentation is clear, but I'm afraid that the motivation of this paper is problematic from the outset. The proposed method cannot reduce the latency of converted SNNs.

**Weaknesses:**

When allowing mixed timesteps across layers, the proposed method enforces sequential dependencies in converted SNNs, where the (i+1) layer must wait for the (i) layer to complete computations for all its timesteps before processing can begin. While current SNNs that can be deployed on neuromorphic hardware actually follow an asynchronous, step-by-step pipeline, this approach changes this paradigm so that the model can only run layer by layer.

Therefore, under such a paradigm shift, the overall inference latency of the converted SNN should correspond to the **SUM** of the timesteps across all layers, rather than their **AVERAGE**. However, as described in Section 5.1, the authors instead report an “average inference time-step", i.e., the averaged layer-wise timesteps as the latency indicator.

Consequently, the latency results reported in this paper are incorrect. The overall delay is not reduced, as the authors claim, but is in fact **increased** due to the sequential execution pipeline that it adopts.

It’s also worth noting that ANN-to-SNN methods published after 2024 will generally include the results on the Transformer architecture, as ResNet/VGG is somewhat outdated. However, this paper does not demonstrate the effectiveness of its approach on the Transformer.

Therefore, I am afraid that the motivation of this paper is problematic from the outset, and I am unable to give a positive rating to the paper.

**Questions:**

See weakness.

---

### Official Review · Reviewer_wxFv · 2025-11-01

**Soundness:** 2
**Presentation:** 3
**Contribution:** 2
**Rating:** 2
**Confidence:** 4

**Summary:**

The paper proposes a mixed-time-step methodology grounded in a "conversion-and-training" paradigm. The method establishes the equivalence relationship between mixed-quantized Artificial Neural Networks (ANNs) and the mixed time-steps of Spiking Neural Networks (SNNs), and further develops a corresponding conversion methodology. Subsequently, the optimal parameter combination is attained through the joint optimization of quantization precision and neurons' firing thresholds. Moreover, the paper rectifies inconsistencies among different time-steps by means of an average firing rate method. Experiments conducted on diverse datasets validate that the proposed methodology delivers performance advantages.

**Strengths:**

- The discussion on loss surfaces across four parameter combinations is insightful. As the loss surface is dominated by the loss function and sample data, a more in-depth theoretical analysis of the optimal parameter combination for optimization is recommended.
- This paper is logically structured and highly accessible.

**Weaknesses:**

- The equivalence between Spiking Neural Networks (SNNs) and Artificial Neural Networks (ANNs) has been discussed, and low-latency conversion methods have been proposed[1].
- Moreover, mixed-timestep SNN conversion approaches have also been put forward[2].
- The average temporal alignment strategy may introduce additional overhead. It is acknowledged that the effort to show the advantages of the proposed average temporal alignment strategy is valuable. However, simulations conducted on GPUs lack persuasiveness. It is recommended that the authors explore the application of neuromorphic hardware.
- A performance comparison with state-of-the-art methods across tasks(such as CIFAR10 with [2]) is encouraged, though this is not deemed critical.

[1]Bu T, Fang W, Ding J, et al. Optimal ANN-SNN conversion for high-accuracy and ultra-low-latency spiking neural networks. ICLR, 2023.
[2]Du K, Wu Y, Deng S, et al. Temporal Flexibility in Spiking Neural Networks: Towards Generalization Across Time Steps and Deployment Friendliness. ICLR, 2025.

Minor:
- line 040~041 '...enabeling enabeling...' duplication
- line 160~161 '...to the resolve inherent temporal dimention mismatch issue...'
- As some papers have been published (such as [1], [2]), please update the references.

**Questions:**

- What's the main difference between the proposed work and [1][2]?
- Could you provide the experimental comparison against [1][2]?

---

### Official Review · Reviewer_eJsg · 2025-11-08

**Soundness:** 3
**Presentation:** 2
**Contribution:** 3
**Rating:** 4
**Confidence:** 5

**Summary:**

This paper proposes Mixed Timestep SNNs, a novel framework that allows different layers to operate with distinct inference timesteps. Within an ANN-to-SNN conversion paradigm, the authors model per-layer timesteps as learnable parameters and jointly optimize them with activation thresholds via end-to-end training. To address inter-layer misalignment caused by heterogeneous timesteps, they introduce an "average temporal alignment" mechanism: the temporal average of upstream spiking outputs is repeated as a static input to downstream layers, effectively decoupling temporal dependencies across layers. Experiments demonstrate that MT-SNNs achieve a state-of-the-art performance for low-latency SNN conversion

**Strengths:**

1.This paper innovatively treats timesteps as differentiable, learnable variables within the ANN-to-SNN conversion framework, moving beyond conventional fixed or heuristic timestep assignments.

2.The method is thoroughly evaluated across multiple architectures and datasets, with comparisons to recent SNN approaches such as QCFS and SRP, yielding convincing accuracy and efficiency results.

**Weaknesses:**

1. Experiments are limited to moderate-scale CNNs. It remains unclear whether the approach generalizes to more complex architectures like Vision Transformers or large-scale models, where heterogeneous timesteps could introduce training instability or convergence issues.

2. The method enforces strictly sequential inter-layer execution: layer l must complete all $T_l$ timesteps before layer l+1 can begin. In contrast, many existing ANN-to-SNN conversion techniques support pipelined or truly asynchronous inference, where layers operate concurrently and total latency scales with the slowest layer rather than the cumulative sum. This fundamental difference in execution models calls into question the fairness of latency–accuracy comparisons reported in the paper.

**Questions:**

1. If layers must be executed sequentially, should the compare the total inference time-step be defined as $\sum_l T_l$ with other works rather than the reported average time-step? Please clarify the time-step used throughout the evaluation.

---

### Note · Authors · 2026-01-08

I have read and agree with the venue's withdrawal policy on behalf of myself and my co-authors.